# Dependency Structure Search Bayesian Optimization for Decision Making Models

**Mohit Rajpal**                                    *mohitr@comp.nus.edu.sg*
*Department of Computer Science*
*National University of Singapore*

**Lac Gia Tran**                                    *tranlac@comp.nus.edu.sg*
*Department of Computer Science*
*National University of Singapore*

**Yehong Zhang**                                    *zhangyh02@pcl.ac.cn*
*Peng Cheng Lab*

**Bryan Kian Hsiang Low**                           *lowkh@comp.nus.edu.sg*
*Department of Computer Science*
*National University of Singapore*

**Reviewed on OpenReview:** `https://openreview.net/forum?id=U6bA2lhwVV`

## Abstract

Many approaches for optimizing decision making models rely on gradient based methods requiring informative feedback from the environment. However, in the case where such feedback is sparse or uninformative, such approaches may result in poor performance. Derivative-free approaches such as Bayesian Optimization mitigate the dependency on the quality of gradient feedback, but are known to scale poorly in the high-dimension setting of complex decision making models. This problem is exacerbated if the model requires interactions between several agents cooperating to accomplish a shared goal. To address the dimensionality challenge, we propose a compact multi-layered architecture modeling the dynamics of agent interactions through the concept of role. We introduce Dependency Structure Search Bayesian Optimization to efficiently optimize the multi-layered architecture parameterized by a large number of parameters, and show an improved regret bound. Our approach shows strong empirical results under malformed or sparse reward.

## 1 Introduction

Decision Making Models choose sequences of actions to accomplish a goal. Multi-Agent Decision Making Models choose actions for multiple agents working together towards a shared goal. Multi-Agent Reinforcement Learning (MARL) has emerged as a competitive approach for optimizing Decision Making Models in the multi-agent setting.[1] MARL optimizes a *policy* under the partially observable Markov Decision Process (POMDP) framework, where decision making happens in an *environment* determined by a set of possible states and actions, and the *reward* for an action is conditioned upon the partially observable state of the environment. A policy forms a set of decision-making rules capturing the most rewarding actions in a given state. MARL utilizes gradient-based methods requiring informative gradients to make progress. This approach benefits from dense reward, which allows reinforcement learning methods to infer a causal relationship between individual actions and their corresponding reward. This feedback may not be present in the scenario of sparse

---

[1]We include an overview of approaches in Decision Making Models in Section 3.

reward(Pathak et al., 2017; Qian & Yu, 2021). In addition, gradient-based methods are susceptible to falling into local maxima.

In contrast to optimization by MARL, Bayesian Optimization (BO) offers an alternative approach to policy optimization. Since BO is a gradient-free optimizer capable of searching globally, applying BO to multi-agent policy search (MAPS) both ensures global searching of the policy, and overcomes poor gradient behavior in the reward function (Qian & Yu, 2021). The chief challenge in BO for MAPS is the high dimensionality of complex multi-agent interactions.

A significant degree of high-dimensional multi-agent interactions exist in MAPS. For example, considering an autonomous drone delivery system, several agents (i.e., drones) must work together to maximize the throughput of deliveries. In doing so, these agents may separate themselves into different roles, for example, long-distance or short-distance deliveries. The optimal policy for each role may be significantly different due to distances to recharging base stations (e.g., drones must conserve battery). In forming the optimal policy, the *interaction* between agents must be considered to both optimally divide the task between the drones, as well as coordinate actions between drones (e.g., collision avoidance). These interactions may change over time. For example, a drone must avoid collision with nearby drones, which changes as it moves through the environment. With many agents, these interactions become more complex.

However, we propose the usage of BO for MAPS on memory-constrained devices which necessitates very compact policies which enables the possibility of overcoming the above limitation. In the context of memory-constrained devices such as Internet of Things (IoT) devices (Merenda et al., 2020), small policies must be used. Secondly, in environments with sparse reward feedback, training these networks with RL presents significant challenges due to unhelpful policy gradients. Finally, the possibility of *globally optimizing* a compact policy for memory-constrained systems is appealing due to its strong performance guarantees.

To allow for the construction of compact policies, we utilize specific multi-agent abstractions of *role* and *role interaction.* In role-based multi-agent interactions, an agent's policy depends on its current role and sparse interactions with other agents. By simplifying the policy space with these abstractions, we increase its tractability for global optimization by BO and inherit the strong empirical performance demonstrated by these approaches. We realize this simplification of the policy space by expressing the role abstraction and role interaction abstractions as immutable portions of the policy space, which are not searched over during policy optimization. To achieve this, we use a higher-order model (HOM) which *generates* a policy model. The HOM is divided into immutable instructions (i.e., algorithms) corresponding to the abstractions of the role and role interaction and mutable parameters that are used to generate (GEN) a policy model during evaluation.

To optimize our proposed HOM, we specialize BO by exploiting task-specific structures. A promising avenue of High-dimensional Bayesian Optimization (HDBO) is through additive decomposition. Additive decomposition separates a high-dimensional optimization problem into several independent low-dimensional sub-problems (Duvenaud et al., 2011; Kandasamy et al., 2015). These sub-problems are independently solved thus reducing the complexity of high dimensional optimization. However, a significant challenge in additive decomposition is *learning the independence structure* which is unknown a-priori. Learning the additive decomposition is accomplished using stochastic sampling such as Gibbs sampling (Kandasamy et al., 2015; Rolland et al., 2018; Han et al., 2020) which is known to have poor performance in high dimensions (Johnson et al., 2013; Barbos et al., 2017).

In our work, we overcome this shortcoming by observing the GEN process of the HOM. In particular, we can measure a surrogate Hessian during the GEN process which significantly simplifies the task of learning the additive structure. This surrogate Hessian informs the dependency structure of the optimization problem due to the equivalence between a zero Hessian value, and independence between dimensions due to the linearity of addition. We term this approach Dependency Structure Search GP-UCB (DSS-GP-UCB) and visualize our approach in Fig. 2. Our proposed BO approach is also applicable to policy-search in the single-agent setting, showing its general-purpose applicability in Decision Making Models. In this work, we make the following contributions:

- We propose a parameter-efficient HOM for MAPS which is both expressive and compact. Our approach is made feasible by using specific abstractions of *roles* and *role interactions.*

- We propose DSS-GP-UCB, a variant of BO that simplifies the learning of dependency structure and provides strong regret guarantees which scale with $\mathcal{O}(\log(D))$ under reasonable assumptions.

- We validate our approach on several multi-agent benchmarks and show our approach outperforms related works for compact models fit for memory-constrained scenarios. Our DSS-GP-UCB also overcomes sparse reward behavior in the reward function in multiple settings showing its effectiveness in Decision Making Models both in the single-agent and multi-agent settings.

## 2 Background

**Bayesian Optimization:** Bayesian optimization (BO) involves sequentially maximizing an unknown objective function $v : \Theta \to \mathbb{R}$. In each iteration $t = 1, \ldots, T$, an input query $\theta_t$ is evaluated to yield a noisy observation $y_t \triangleq v(\theta_t) + \epsilon$ with i. i. d. Gaussian noise $\epsilon \sim \mathcal{N}(0, \sigma^2)$. BO selects input queries to approach the global maximizer $\theta^* \triangleq \arg\max_{\theta \in \Theta} v(\theta)$ as rapidly as possible. This is achieved by minimizing *cumulative* regret $R_T \triangleq \sum_{t=1}^{T} r(\theta_t)$, where $r(\theta_t) \triangleq v(\theta^*) - v(\theta_t)$. Cumulative regret is a key performance metric of BO methods.

The probability distribution of $v$ is modeled by a *Gaussian process* (GP), denoted GP $(\mu(\theta), k(\theta, \theta'))$, that is, every finite subset of $\{v(\theta)\}_{\theta \in \Theta}$ follows a multivariate Gaussian distribution (Rasmussen & Williams, 2006). A GP is fully specified by its *prior* mean $\mu(\theta)$ and covariance $k(\theta, \theta')$ for all $\theta, \theta' \in \Theta$, which are, respectively, assumed w.l.o.g. to be $\mu(\theta) = 0$ and $k(\theta, \theta') \leq 1$. Given a vector $\mathbf{y}_T \triangleq [y_t]_{t=1,\ldots,T}^{\top}$ of noisy observations from evaluating $v$ at input queries $\theta_1, \ldots, \theta_T \in \Theta$ after $T$ iterations, the GP posterior probability distribution of $v$ at some input $\theta \in \Theta$ is a Gaussian with the following *posterior* mean $\mu_T^k(\theta)$ and variance $[\sigma_T^k]^2(\theta)$:

$$\mu_T^k(\theta) \triangleq \mathbf{k}_T^k(\theta)^{\top}(\mathbf{K}_T^k + \sigma^2\mathbf{I})^{-1}\mathbf{y}_T, \quad \left[\sigma_T^k\right]^2(\theta) \triangleq k(\theta, \theta) - \mathbf{k}_T^k(\theta)^{\top}(\mathbf{K}_T^k + \sigma^2\mathbf{I})^{-1}\mathbf{k}_T^k(\theta) \tag{1}$$

where $\mathbf{K}_T^k \triangleq [k(\theta_t, \theta_{t'})]_{t,t'=1,\ldots,T}$ and $\mathbf{k}_T^k(\theta) \triangleq [k(\theta_t, \theta)]_{t=1,\ldots,T}^{\top}$. In each iteration $t$ of BO, an input query $\theta_t \in \Theta$ is selected to maximize the GP-UCB acquisition function, $\theta_t \triangleq \arg\max_{\theta \in \Theta} \mu_{t-1}(\theta) + \sqrt{\beta_t}\sigma_{t-1}(\theta)$ (Srinivas et al., 2010) where $\beta_t$ follows a well defined pattern.

## 3 Related work

**Decision Making Models:** Decision Making Models (Rizk et al., 2018; Roijers et al., 2013) determine actions taken by an agent or agents in order to achieve a goal. We focus on the POMDP setting and optimizing a policy to accumulate maximum reward while interacting with a partially observable environment (Shani et al., 2013). Many approaches exist which can be broadly categorized into direct policy search and reinforcement learning methods. Direct policy search (Heidrich-Meisner & Igel, 2008; Lizotte et al., 2007; Martinez-Cantin, 2017; Papavasileiou et al., 2021; Wierstra et al., 2008) searches the policy space in some efficient manner. Reinforcement learning (Arulkumaran et al., 2017; Fujimoto et al., 2018; Haarnoja et al., 2018; Lillicrap et al., 2015; Lowe et al., 2017; Mnih et al., 2015; Schulman et al., 2017) starts with a randomly initialized policy and *reinforces* rewarding behavior patterns to improve the policy.

**Bayesian Optimization for Decision Making Models:** BO has been utilized for direct policy search in the low dimensional setting (Lizotte et al., 2007; Wilson et al., 2014; Marco et al., 2016; Martinez-Cantin, 2017; von Rohr et al., 2018). However, these approaches have not scaled to the high dimensional setting. In more recent works, BO has been utilized to aid in local search methods similar to reinforcement learning (Akrour et al., 2017; Eriksson et al., 2019a; Wang et al., 2020a; Fröhlich et al., 2021; Müller et al., 2021). However, these approaches require evaluation of an inordinate number of policies typical of local search methods and do not provide regret guarantees. Recently, combinations of local and global search methods have been proposed (McLeod et al., 2018; Shekhar & Javidi, 2021). However, these approaches rely on informative and useful gradient information and have not been shown to scale to the high dimensional setting.

**MARL for multi-agent decision making:** A well-known approach for cooperative MARL is a combination of centralized training and decentralized execution (CTDE) (Oliehoek et al., 2008). The multi-agent interactions of CTDE methods can be implicitly captured by learning approximate models of other agents (Lowe et al., 2017; Foerster et al., 2018) or decomposing global rewards (Sunehag et al., 2017; Rashid et al.,

2018; Son et al., 2019). However, these methods do not focus on how interactions are performed between agents. In MARL, the concept of *role* is often leveraged to enhance the flexibility of behavioral representation while controlling the complexity of the design of agents (Lhaksmana et al., 2018; Wang et al., 2020b; 2021b; Li et al., 2021). Our approach is related to the study of (Le et al., 2017) where the interactions are also captured by role assignment. However, the approach operates on an imitation learning scenario, and the role assignment depends on the heuristic from domain knowledge. Another related field is Comm-MARL (Zhu et al., 2022; Shao et al., 2022; Liu et al., 2020; Peng et al., 2017; Das et al., 2019; Singh et al., 2019), where agents are allowed to communicate during policy execution to jointly decide on an action. In contrast, our approach utilizes both abstractions of role and role interaction in a HOM for a decision making model.

## 4 Design

We consider the problem of learning the joint policy of a set of $n$ agents working cooperatively to solve a common task. During each interaction with the environment, each agent $i$ is associated with a state $\mathbf{s}^i \in \mathcal{S}^i$ with the global state represented as $\mathbf{s} \triangleq [\mathbf{s}^i]_{i=1,\ldots,n}$. Each agent $i$ cooperatively chooses an action $\mathbf{a}^i \in \mathcal{A}^i$ with the global action represented by $\mathbf{a} \triangleq [\mathbf{a}^i]_{i=1,\ldots,n}$. Each state, action pair is associated with a *reward* function: $\rho(\mathbf{s}, \mathbf{a})$. In order to achieve the common task, a policy parameterized by $\theta$: $\pi^\theta \triangleq \mathcal{S} \to \mathcal{A}$ governs the action taken by the agents, after observing state $\mathbf{s} \in \mathcal{S}$. The goal of RL is to learn the optimal policy parameters that maximizes the accumulation of rewards during a predefined number of interactions with the environment,[2] $v(\theta)$. In contrast to RL, which receives feedback on the reward of an action with every interaction, we treat $v(\theta)$ as an opaque function measuring the *value* of a policy. We utilize BO to optimize $\theta$ using solely the accumulated reward, $v(\theta)$, as feedback from the environment.

### 4.1 Architectural design

To achieve a compact and tractable policy space, we consider policies under the useful abstractions of *role* and *role interaction*. These abstractions have consistently shown strong performance in multi-agent tasks. Therefore, we can simplify the policy space by limiting it to only policies using these abstractions, but still have powerful and expressive policies suitable for multi-agent systems.

As role and role interaction are immutable abstractions within our policy space, we express them as *static algorithms* which are not searched over during policy optimization. These algorithms take as input parameters which are mutable and searched over during policy optimization. This combination of immutable instructions, and mutable parameters reduces the size of the search space,[3] yet is still able to express policies which conform to the role and role interaction abstractions.

We term this approach a *higher-order model* (HOM) which generates (GEN) the model using instructions and parameters into a policy model during evaluation. This HOM is separated into role assignment, and role interaction stages. We visualize an overview of this approach in Fig. 1, left. The HOM parameters are interpreted in context of the current state by the instructions (Alg. 1, Alg. 2, Alg. 3) of the HOM to form the policy model which dictates the resultant action. In our work, each HOM component of role assignment and role interaction is implemented as a neural network.

### 4.2 Role assignment

Following the success of role based collaboration in multi-agent systems, we assume the interaction and decision making of each agent is governed by its assigned role. For example, in drone delivery, roles could be short-distance deliveries, and long-distance deliveries. In filling these roles, the state of each of the agents are considered. E.g., a drone with low battery may be limited to only performing short-distance deliveries. A straightforward approach to implement role based interaction is to permute agents into an equivalent number of roles.[4] We assume that an optimal policy can be decomposed as follows:

---

[2]Further RL overview can be found in Arulkumaran et al. (2017).
[3]This approach to efficiency is similar in spirit to the work of Lee et al. (1986).
[4]This is a common assumption in multi-agent systems, see, e.g., Le et al. (2017).

$$\pi(\mathbf{a}^1, \ldots, \mathbf{a}^n \mid \mathbf{s}^1, \ldots, \mathbf{s}^n) \triangleq \pi_r(\mathbf{a}^{\alpha(1)}, \ldots, \mathbf{a}^{\alpha(n)} \mid \mathbf{s}^{\alpha(1)}, \ldots \mathbf{s}^{\alpha(n)}) \tag{2}$$

where $\alpha$ is a permutation function dependent on the state, $\mathbf{s}^1, \ldots, \mathbf{s}^n$. Our approach to role assignment is simple and general purpose, which is also well studied and theoretically principled.

To capture this behavior, we utilize a per role affinity function: $\Lambda^{\theta_{r,i}}(\cdot)$ which is the affinity to take on role $i$ and is parameterized by $\theta_{r,i}$. This function evaluates the affinity of agent $\ell$ taking on role $i$ using the state of agent: $\mathbf{s}^\ell$. The optimal permutation maximizes the total affinity of an assignment: $\sum_{i=1}^n \Lambda^{\theta_{r,i}}(\mathbf{s}^{\alpha(i)})$ where $\alpha$ represents a permutation. This problem can be efficiently solved using the Hungarian algorithm (Kuhn, 1955). We integrate the Hungarian algorithm in our HOM approach during the GEN process. We formalize this in Algorithm 1 which forms the instructions in the role assignment HOM.

Given Algorithm 1, during GEN process, the agents' state, $\mathbf{s}^1, \ldots, \mathbf{s}^n$ is contextually interpreted to yield a permutation model: $\alpha$. Going forward, we consider the problem of determining the joint policy $\pi_r(\mathbf{a}^{\alpha(1)}, \ldots, \mathbf{a}^{\alpha(n)} \mid \mathbf{s}^{\alpha(1)}, \ldots \mathbf{s}^{\alpha(n)})$ which enables collaborative interactions.

### 4.3 Role interaction

Capturing multiple roles working together is an important part of an effective multi-agent policy. For example in drone delivery, drones must both divide the available task among themselves, as well as use collision avoidance while executing deliveries. Modeling role interactions must accomplish two goals. Firstly, agent interactions may change over time. For example collision avoidance strategies involve the closest drones which change as the drone moves within the environment. Secondly, efficient parameterization is needed as the number of interactions can scale exponentially due to considering interactions between many agents.

To overcome these challenges, we propose a HOM which generates (GEN) a graphical model. The usage of a graphical model decomposes the exponentially scaling interaction problem into a pairwise interaction model, along with a message passing approach to facilitate complex interactions between many agents.[5] The GEN process is conditioned on the agents' state, thus enabling dynamic role interactions; in addition the GEN process allows for a more compact policy space with far fewer parameters. The resultant generated graphical model captures the state-dependent interaction between roles and yields the resultant actions for each role. After GEN, the interaction between roles are captured by the resultant conditional random field. This is presented in Fig. 1, right. The MRF (Markov Random Field)

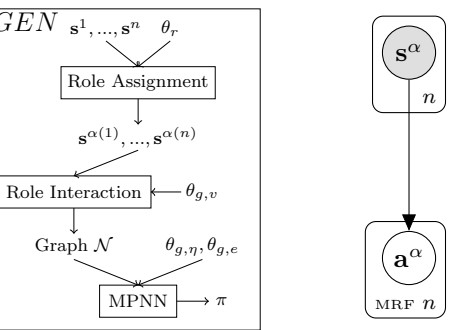

Figure 1: Left: HOM architecture. GEN uses $\theta_r$ and $\theta_g$ during evaluation to yield a model which represents the policy. $\theta_r$ and $\theta_g$ are optimized by BO. Right: Inferring $\mathbf{a}^\alpha$ given $\mathbf{s}^\alpha$.

represents arbitrary undirected connectivity between nodes $\mathbf{a}^{\alpha(1)}, \ldots, \mathbf{a}^{\alpha(n)}$, which is denoted by $\mathcal{G}$. This connectivity allows different roles to collaborate together to determine the joint action. To generate graphical models of the above form, our HOM uses edge affinity functions, $\Lambda^{\theta_{g,v}}(\cdot)$, which enables dynamic arbitrary connectivity between roles. For all pairs of roles with state, $\mathbf{s}^{\alpha(i)}, \mathbf{s}^{\alpha(\ell)}$ an edge is generated if the affinity between these two states is sufficiently high (i.e., $> 0$). This dynamic edge generation approach overcomes the quadratic parameter scaling if all pairs of agents were separately modelled. The graphical model GEN process is presented in Algorithm 2 which yields a graphical model.

To cooperatively determine a set of actions for roles given the graphical model, we perform inference over the graphical model presented in Fig. 1 using Message Passing Neural Networks (Gilmer et al., 2017) (MPNN). We present iterative message passing rules to map from $\mathbf{s}^\alpha$ to $a^\alpha$:

---

[5]We refer readers to Wang et al. (2013) for additional overview on graphical models.

$$m_{t+1}^{\alpha(i)} \triangleq \sum_{\alpha(\ell) \in N^{\alpha(i)}} M^{\theta_{g,\eta}}\left(h_t^{\alpha(i)}, h_t^{\alpha(\ell)}, i, \ell\right); \quad h_{t+1}^{\alpha(i)} \triangleq U^{\theta_{g,e}}\left(\mathbf{s}^{\alpha(\mathbf{i})}, h_t^{\alpha(i)}, m_{t+1}^{\alpha(i)}\right); \quad \mathbf{a}^{\alpha} \triangleq \left[h_\tau^{\alpha(i)}\right]_{i=1,\ldots,n} \tag{3}$$

where $M$ is the message function parameterized by $\theta_{g,\eta}$ which enables interaction between connected nodes, $U$ is the action update function parameterized by $\theta_{g,e}$ which updates the node's internal hidden state conditioned on the messages received, and $N^{\alpha(i)}$ denotes the neighbors of $\alpha(i)$. The message passing procedure allows for cooperative determination of all roles' actions using pairwise message passing. Roles which are not immediate neighbors of each other influence each other's behavior through intermediary connecting nodes. The message passing procedure concludes after $\tau$ iterations of message passing with the policy actions indicated by the hidden states, $\left[h_\tau^{\alpha(i)}\right]_{i=1,\ldots,n}$.

Finally, Algorithm 3 drives the GEN process. The GEN process consists of permuting agents into roles, creating the graphical model to enable interactions between agents taking on their respective roles, and finally performing inference over the graphical model using a MPNN.

### 4.4 Additive decomposition

Although our HOM policy representation is compact, it is still of significant dimensionality which makes optimization with BO difficult. HDBO is challenging due to the curse of dimensionality with common kernels such as Matern or RBF.[6] This curse of dimensionality stems directly from the difficulty of finding the global optima of a high-dimensional function (e.g., a value function $v(\theta)$ determining the value of a policy in some unknown environment). A common technique to overcome this is through assuming additive structural decomposition on $v$: $v(\theta) \triangleq \sum_{i=1}^{M} v^{(i)}(\theta^{(i)})$ where $v^{(i)}$ are independent functions, and $\theta^{(i)} \in \Theta^{(i)}$ (Duvenaud et al., 2011). The additive decomposition simplifies a high-dimensional optimization problem since the optima of a function constructed through addition of subfunctions can be found by independently optimizing each subfunction as visualized in Fig. 2. In the context of BO, additive decomposition significantly simplifies the optimization problem due to the properties of Multivariate Gaussian variables.

In additive decomposition we denote the domain of the optimization problem, $\Theta \triangleq \Theta^1 \times \ldots \times \Theta^D$ for some dimensionality $D$, that is the domain is constructed through the Cartesian product of each of its dimensions. Each subfunction to optimize, $v^{(i)}$, corresponds to a subdomain restricted to some subset of these dimensions, $\Theta^{(i)} \subseteq \{\Theta^1, \ldots, \Theta^D\}$. Typically, it is assumed that each $\Theta^{(i)}$ is of low dimensionality (i.e., $v^{(i)}$ is defined on only a few dimensions for each $i$). This structural assumption is combined with the assumption that each $v^{(i)}$ is sampled from a GP. Due to the properties of Multivariate Gaussians, if $v^{(i)} \sim \mathrm{GP}\left(0, k^{\Theta^{(i)}}(\theta^{(i)}, \theta^{(i)'})\right)$ then $v \sim \mathrm{GP}\left(0, \sum_i k^{\Theta^{(i)}}(\theta^{(i)}, \theta^{(i)'})\right)$ (Rasmussen & Williams, 2006), which follows from the addition of two Gaussian random variables is also a Gaussian random variable. This assumption decomposes a high dimensional GP surrogate model of $v$ into a set of many low dimensional GPs, which is easier to jointly learn and optimize.

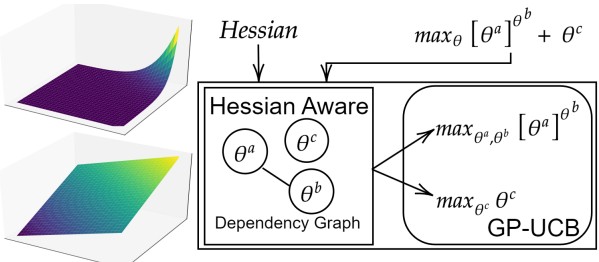

Figure 2: Left, above, plot of $f(x, y) = x^y$; below, plot of $f(x, y) = x + y$. The curvature of *additively constructed functions* is zero; *non-zero curvature* indicates dependency among input variables. Right, examining the Hessian learns the dependency structure which decomposes complex problems into simpler problems solved by GP-UCB.

To contextualize an additive decomposition, we represent the decomposition by a dependency graph between the dimensions: $\mathcal{G}_d \triangleq (V_d, E_d)$ where $V_d \triangleq \{\Theta^1, \ldots, \Theta^D\}$ and $E_d \triangleq \{(\Theta^a, \Theta^b) \mid a, b \in \Theta^{(i)} \text{ for some } i\}$. A simple decomposition of an additive function and its associated dependency graph is visualized in Fig. 2.

---

[6] A parallel area in HDBO is of computational efficiency of acquisition which is outside the scope of this work. We refer readers to the works of Mutny & Krause (2018), Wilson et al. (2020), and Ament & Gomes (2022).

---

**Algorithm 1** *RoleAssignment*

---

**Require:** $\mathbf{s}^1, \ldots, \mathbf{s}^n$
1: **return** $\arg\max_\alpha \sum_{i=1}^n \Lambda^{\theta_r, i}(\mathbf{s}^{\alpha(i)})$

---

**Algorithm 4** DSS-GP-UCB

---

**Require:** $v, H, k$
1: **for** $t \leftarrow 1, \ldots, T_0$ **do**     ▷ Sample Hessian $T_0 \times C_1$ times for dependencies.
2:     $\theta_{t,h} \sim \mathcal{U}(\Theta)$          ▷ Randomly sample over the domain.
3:     **for** $\ell \leftarrow 1, \ldots, C_1$ **do** $h_{t,\ell} \leftarrow H(\theta_{t,h})$
4: $\widetilde{E}_d \leftarrow \left| \sum h \right| > c_h; \widetilde{\mathcal{G}}_d \leftarrow (\{\Theta^1, \ldots, \Theta^D\}, \widetilde{E}_d)$     ▷ Discriminate dependencies
5: $[\Theta^{(i)}]_{i=1,\ldots,M} \leftarrow$ Max-Cliques$(\widetilde{\mathcal{G}}_d); k \leftarrow \sum_{i=1}^M k^{\Theta^{(i)}}$     ▷ Compute Max-Cliques
6: **for** $t \leftarrow T_0, \ldots, T$ **do**          ▷ Run GP-UCB with dependency structure
7:     $\theta_t \leftarrow \arg\max_\theta \mu_{t-1}^k(\theta) + \sqrt{\beta_t}\sigma_{t-1}^k(\theta)$     ▷ Max-Cliques additive kernel
8:     Query $\theta_t$ to observe $y_t = v(\theta_t) + \mathcal{N}(0, \epsilon^2)$
9:     Update posterior, $\mu, \sigma$, with $\theta_t, y_t$

---

**Algorithm 2** *RoleInteraction*

---

**Require:** $\mathbf{s}^{\alpha(1)}, \ldots, \mathbf{s}^{\alpha(n)}$
1: **for** $i \leftarrow 1, \ldots, n$ **do**
2:     **for** $\ell \leftarrow 1, \ldots, n$ **do** ▷ Edge affinities.
3:         **if** $\Lambda^{\theta_g, v}(\mathbf{s}^{\alpha(i)}, \mathbf{s}^{\alpha(\ell)}) > 0$ **then**
4:             $N^{\alpha(i)}.append(\alpha(\ell))$
5: **return** $N^{\alpha(1)}, \ldots, N^{\alpha(n)}$

---

**Algorithm 3** GEN-*Policy*

---

**Require:** $\mathbf{s}^1, \ldots, \mathbf{s}^n$
1: $\alpha \leftarrow RoleAssignment(\mathbf{s}^1, \ldots, \mathbf{s}^n)$
2: $N \leftarrow RoleInteraction(\mathbf{s}^{\alpha(1)}, \ldots, \mathbf{s}^{\alpha(n)})$
3: $\mathbf{a} \leftarrow$ MPNN$(\mathbf{s}^\alpha, N)$          ▷ See Eq. 3
4: **return** $[a^{\alpha^{-1}(i)}]_{i=1,\ldots,n}$

---

*We **highlight** that this graph is between the dimensions of the policy parameters, $\Theta$, and is unrelated to the graphical model of role interactions presented in earlier sections. It is possible to accurately model $v$ by a kernel $k \triangleq \sum_i k^{\Theta^{(i)}}$ where each $\Theta^{(i)}$ corresponds to a maximal clique of the dependency graph* (Rolland et al., 2018). *Knowing the dependency graph greatly simplifies the complexity of optimizing $v$.*

However, learning the dependency graph in additive decomposition remains challenging as there are $\mathcal{O}(D^2)$ possible edges each of which may be present or absent yielding $2^{\mathcal{O}(D^2)}$ possible dependency structures. This difficult problem is often approached using inefficient stochastic sampling methods such as Gibbs sampling.

### 4.5 Dependency Structure Search Bayesian Optimization

We propose learning the dependency structure during the GEN process. Our proposed approach is based on the following observation, which is illustrated in Fig. 2.

**Proposition 1.** *Let $\mathcal{G}_d = (V_d, E_d)$ represent an additive dependency structure with respect to $v(\theta)$, then the following holds true: $\forall a, b \; \frac{\partial^2 v}{\partial \theta^a \partial \theta^b} \neq 0 \implies (\Theta^a, \Theta^b) \in E_d$ which is a consequence of $v$ formed through addition of independent sub-functions $v^{(i)}$, at least one of which must contain $\theta^a, \theta^b$ as parameters for $\frac{\partial^2 v}{\partial \theta^a \partial \theta^b} \neq 0$ which implies their connectivity within $E_d$.*

In practice, observing the Hessian of the value function, $\mathbf{H}_v$, is not possible due to $v$ being an opaque function. However, during the GEN process we can observe the Hessian of the policy, $\mathbf{H}_\pi$. This surrogate Hessian is closely related to the $\mathbf{H}_v$ as $v(\theta)$ is determined through interaction of the policy with an unknown environment. Because the *value* of a policy is a function of the policy; it follows by the chain rule that $\mathbf{H}_\pi$ is an important sub-component of $\mathbf{H}_v$. We utilize the surrogate Hessian in our work and demonstrate its strong empirical performance in validation. Following this reasoning, we consider algorithms with noisy query access to the Hessian, $\mathbf{H}_v$. Note that we assume that the surrogate Hessian, $\mathbf{H}_\pi$, can well serve as a noisy surrogate for the true Hessian, $\mathbf{H}_v$.[7]

**Assumption 1.** *Let $\mathcal{G}_d = (V_d, E_d)$ be sampled from an Erdős-Rényi model with probability $p_g < 1$: $\mathcal{G}_d \sim G(D, p_g)$. That is, each edge $(\Theta^a, \Theta^b)$ is i.i.d. sampled from a binomial distribution with probability, $p_g$. With $[\Theta^{(i)}]_{i=1,\ldots,M}$ representing the maximal cliques of $\mathcal{G}_d$, we assume that $v \sim GP\left(0, \sum_i k^{\Theta^{(i)}}(\theta^{(i)}, \theta^{(i)'})\right)$ for some kernel $k$ taking an arbitrary number of arguments (e.g., RBF). Noisy queries can be made to the Hessian of $v$, $\mathbf{H}_v$. We define $H(\theta) \triangleq [\frac{\partial^2 v}{\partial \theta^a \partial \theta^b} + \epsilon_h^{(a,b)}]_{a,b=1,\ldots,D}$ where $\epsilon_h^{(a,b)} \sim \mathcal{N}(0, \sigma_n^2)$ i.i.d. Each query to $H$ has corresponding regret of $r(\theta)$.*

Under this assumption, we show that it's possible to learn the underlying dependency structure of $\mathcal{G}_d = (V_d, E_d)$ with a polynomial number of queries to the noisy Hessian. We present DSS-GP-UCB in Algorithm 4 and prove theoretical results regarding its performance. In the first stage of DSS-GP-UCB, we perform $C_1$ queries to the

---

[7]We revisit the validity of this assumption in Appendix H.

Hessian if $t \leq T_0$. These Hessian queries are then averaged and compared to a cutoff constant $c_h$ to determine the dependency structure $\widetilde{E}_d$. We show that after $C_1 T_0$ queries to the Hessian, with high probability we have $\widetilde{E}_d = E_d$, where $E_d$ is the unknown ground truth dependency structure for $v$. This argument is formalized in the following theorem.

**Theorem 1.** *Suppose[8] there exists $\sigma_h^2, p_h$ s.t. $\forall i,j$ $\mathbb{P}_{\theta \sim \mathcal{U}(\Theta)} \left[ k^{\partial i \partial j}(\theta, \theta) \geq \sigma_h^2 \right] \geq p_h$ and $\forall i,j,\theta,\theta'$ $k^{\partial i \partial j}(\theta, \theta') \geq 0$. Then for any $\delta_1, \delta_2 \in (0,1)$ after $t \geq T_0$ steps of DSS-GP-UCB we have: $\bigcap_{i,j} P(\widetilde{E}_d^{i,j} = E_d^{i,j}) \geq 1 - \delta_1 - \delta_2$ when $T_0 = C_1 > \frac{16D^2}{p_h \delta_1^2} \log \frac{2D^2}{\delta_1} \frac{\sigma_n^2}{\sigma_h^2} + \frac{D^2}{2\delta_2}$, $c_h \triangleq T_0 \sigma_n \sqrt{2 \log \frac{2D^2}{\delta_1}}$.*

Our Theorem 1 relies on repeatedly sampling the Hessian to determine whether an edge exists between $\Theta^a$, and $\Theta^b$ in the sampled additive decomposition. The key challenge is determining this connectivity under a very noisy setting, and for extremely low values of $\sigma_h^2 \ll \sigma_n^2$ where the Hessian is zero with high probability. We are able to overcome this challenge using a Bienaymé's identity, a key tool in our analysis. We defer all proofs to the Appendix.

In the second stage of DSS-GP-UCB, we extract the maximal cliques depending on $\widetilde{E}_d$ and construct the GP kernel, $k = \sum_i k^{\Theta^{(i)}}$, the sum of the aforementioned kernels and inference and acquisition proceeds same as GP-UCB (lines 6-9).

To bound the cumulative regret, $R_t \triangleq \sum_{t=1}^{T_0} C_1 r(\theta_{t,h}) + \sum_{t=T_0}^{T} r(\theta_t)$, we follow the following process. First, we bound the number and size of cliques of graphs sampled from the Erdős-Rényi model with high probability. Second, we bound the *mutual information* of an additive decomposition given the mutual information of its constituent kernels using Weyl's inequality. Third, we use similar analysis as Srinivas et al. (2010) to complete the regret bound.

**Theorem 2.** *Let $k$ be the kernel as in Assumption 1, and Theorem 1. Let $\gamma_T^k(d) : \mathbb{N} \to \mathbb{R}$ be a monotonically increasing upper bound function on the* mutual information *of kernel $k$ taking $d$ arguments. The cumulative regret of DSS-GP-UCB is bounded with high probability as follows:*

$$R_T = \widetilde{\mathcal{O}}\left( \sqrt{T \beta_T D^{\log D + 5} \gamma_T^k (4 \log D + c_\gamma)} \right) \tag{4}$$

*where $c_\gamma$ is an appropriately picked constant and the base of the logarithm is $\frac{1}{p_g}$.*

Whereas for typical kernels such as Matern and RBF, cumulative regret of GP-UCB scales exponentially with $D$, our regret bounds scale with exponent $\mathcal{O}(\log D)$. This improved regret bound shows our approach is a theoretically grounded approach to HDBO.

# 5 Validation

We compare our work against recent algorithms in MARL on several multi-agent coordination tasks and RL algorithms for policy search in novel settings. We also perform ablation and investigation of our proposed HOM at learning roles and multi-agent interactions. We defer experimental details to Appendix A.

*All presented figures are average of 5 runs with shading representing $\pm$ Standard Error, the y-axis represents cumulative reward, the x-axis displayed above represents interactions with the environment in RL, x-axis displayed below represents iterations of BO. Commensurate with our focus on memory-constrained devices, all policy models consist of $< 500$ parameters.*

## 5.1 Ablation

We investigate the impact of Role Assignment (RA) and Role Interaction (RI) as well as model capacity on training progress. We conduct ablation experiments on Multiagent Ant with 6 agents, PredPrey with 3 agents, and Heterogenous PredPrey with 3 agents (Peng et al., 2021). Multiagent Ant is a MuJoCo (Todorov et al., 2012) locomotion task where each agent controls an individual appendage. PredPrey is a task where

---

[8]RBF kernel satisfies these assumptions when $\Theta = [0,1]^D$.

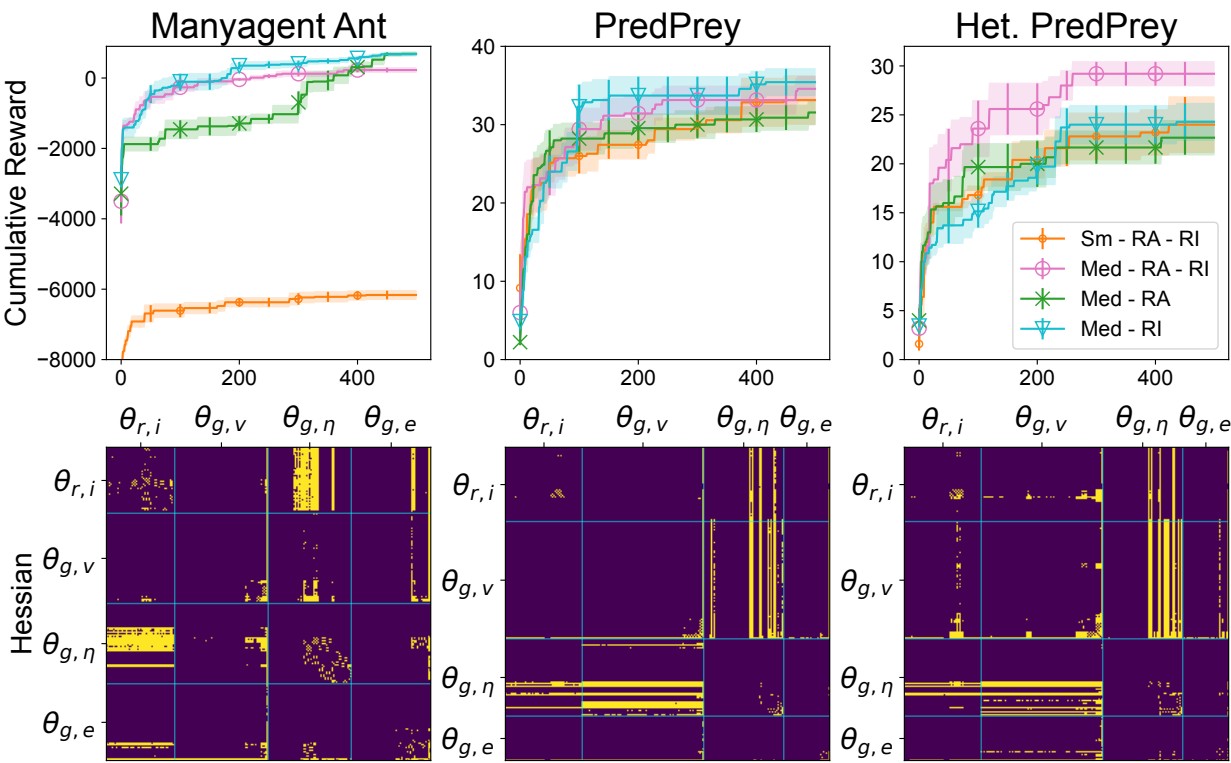

Figure 3: Ablation study. Training curves of our HOM and its ablated variants on different multi-agent environments.

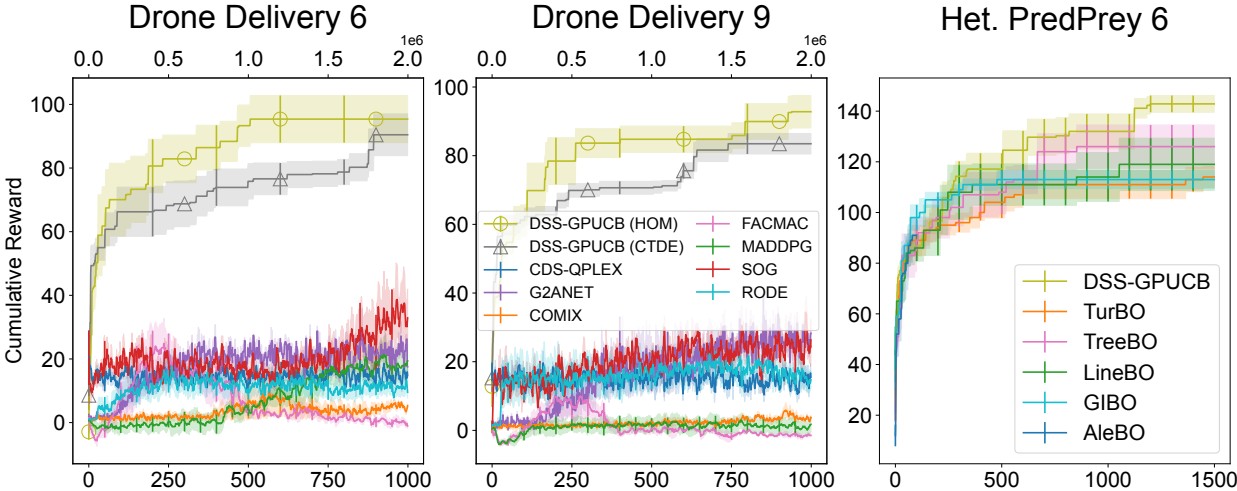

Figure 4: Left two plots: Sparse reward drone delivery task. Rightmost: Comparison with HDBO approaches. The left two plots validate the same approaches on different environments.

predators must work together to catch faster, more agile prey. Het. PredPrey is similar, except the predators have different capabilities of speed and acceleration. In ablation experiments, our default configuration is *Med - RA - RI* which employs components of RA and RI parameterized by neural networks with three layers and four neurons on each layer (medium sized neural network). The *Sm*, small, model is instead parameterized with neural networks of 1 layer with 2 neurons each. When RA is ablated, the agents interact directly without taking on any role based specialization. When RI is ablated, the agents' action is determined without any coordination between agents. We present our ablation in Fig. 3.

Table 1: DSS-GP-UCB typically outperforms RL with higher sparsity (e.g., Sparse-100, or Sparse-200).

| | Ant-v3 | | | | | Hopper-v3 | | | | | Swimmer-v3 | | | | | Walker2d-v3 | | | | |
|---|---|---|---|---|---|---|---|---|---|---|---|---|---|---|---|---|---|---|---|---|
| | DDPG | PPO | SAC | TD3 | Intrinsic | DDPG | PPO | SAC | TD3 | Intrinsic | DDPG | PPO | SAC | TD3 | Intrinsic | DDPG | PPO | SAC | TD3 | Intrinsic |
| Baseline | −90.77 | 1105.69 | 2045.24 | 2606.17 | 2144.00 | 604.20 | 1760.65 | 2775.66 | 1895.76 | 1734.00 | 44.45 | 121.38 | 58.73 | 48.78 | 1950.00 | 2203.80 | 892.81 | 4297.03 | 1664.46 | 2210.00 |
| Sparse 2 | −32.88 | 1007.80 | 2563.97 | 1407.40 | 1964.00 | 877.93 | 1567.14 | 3380.60 | 1570.84 | 2074.00 | 35.59 | 99.50 | 46.75 | 47.23 | 1758.80 | 1470.62 | 1471.33 | 1673.46 | 2297.43 | 1952.00 |
| Sparse 5 | −2687.97 | 961.31 | 711.56 | 762.61 | 1916.00 | 814.59 | 1616.79 | 3239.20 | 2290.67 | 1972.00 | 26.66 | 68.69 | 43.84 | 40.12 | 1856.00 | 961.30 | 697.93 | 1697.25 | 2932.27 | 1924.00 |
| Sparse 20 | −2809.89 | 624.07 | 694.30 | 379.12 | 1838.00 | 783.95 | 1629.28 | 2535.17 | 1436.33 | 1537.20 | 19.12 | 54.63 | 37.78 | 37.03 | 2108.00 | 663.04 | 365.39 | 1010.63 | 276.56 | 1810.00 |
| Sparse 50 | −3067.37 | −67.43 | 663.28 | 253.66 | 1091.20 | 816.25 | 1010.73 | 1238.03 | 551.43 | 642.00 | 23.73 | 51.52 | 38.78 | 30.01 | 812.00 | 572.12 | 428.29 | 349.47 | 298.28 | 834.75 |
| Sparse 100 | −3323.43 | −4021.56 | 679.30 | −115.43 | 450.40 | 988.36 | 324.51 | 260.52 | 342.48 | 406.80 | 9.64 | 21.09 | 27.98 | 30.10 | 376.60 | 523.89 | 205.93 | 200.16 | 147.22 | 480.60 |
| Sparse 200 | −3098.37 | −8167.98 | −107.14 | −147.86 | 258.60 | 765.05 | 222.76 | 300.36 | 281.68 | 350.80 | −9.97 | 21.69 | 33.35 | 30.48 | 342.80 | 182.84 | 193.43 | 187.16 | 148.06 | 353.20 |
| DSS-GP-UCB | 1147.21 | | | | | 1009.3 | | | | | 175.73 | | | | | 1008.90 | | | | |

For a simpler coordination task such as Multiagent Ant, we observe limited improvement through RA or RI. In contrast, RI shows strong improvement in PredPrey and Het. PredPrey. It is because, in PredPrey, predators must work together to catch the faster prey. Since the agents in PredPrey are *homogeneous*, *ablating RA* makes the optimization simpler and more compact without losing expressiveness. Thus, ablating RA leads to a performance increase. In Het. PredPrey, the predator agents have heterogeneous capabilities in speed and acceleration. Thus, RA plays a critical role in delivering strong performance. We also show that overly shrinking the model size (*Sm - RA - RI*) can hurt performance as the policy model is no longer sufficiently expressive. This is evidenced in the Multiagent Ant task. We observed that using neural networks of three layers with four neurons each to be sufficiently balanced across a wide variety of tasks.

In Fig. 3, we present the detected Hessian structure by DSS-GP-UCB in the respective tasks. The detected Hessian structures generally show strong block-diagonal associativity in the HOM parameters, i.e., $[\theta_{r,i}, \theta_{g,v}, \theta_{g,\eta}, \theta_{g,e}]$. This shows that our approach can detect the interdependence *within* the sub-parameters, but relative independence between the sub-parameters. We observe more off-diagonal connectivity in the complex coordination tasks of PredPrey and Het. PredPrey. The visualization of Hessian structure on PredPrey shows that our approach can detect the importance of *jointly optimizing* role assignment and interaction to deliver a strong policy in this complex coordination task. We investigate the learning behavior of the HOM further in Appendix B.

## 5.2 Comparison with MARL

We compare our method with competing MARL algorithms on several multi-agent tasks where the number of agents is increased. We validate both the HOM with DSS-GP-UCB (DSS-GP-UCB (MM)) and neural network policies trained in the CTDE paradigm (DSS-GP-UCB (CTDE)). In the CTDE paradigm, both RI and RA are ablated reducing the policy model to a neural network which is identical across all agents. We observe that on complex coordination tasks such as PredPrey and Het. PredPrey our approach delivers more performant policies when coordination is required between *a large number of agents*. This is presented[9] in Fig. 5. Although SOG (Shao et al., 2022), a Comm-MARL approach shows compelling performance with a small number of agents, with 15 agents, both DSS-GP-UCB (CTDE) and DSS-GP-UCB (MM) outperform this strategy. We highlight that DSS-GP-UCB (CTDE) outperforms Comm-MARL approaches without communication during execution. We also note that DSS-GP-UCB (MM) outperforms DSS-GP-UCB (CTDE) showing the value of our HOM approach in complex coordination tasks. We defer further experimental results in this setting to Appendix B.

## 5.3 Policy optimization under malformed reward

We compare against several competing RL and MARL algorithms under malformed reward scenarios. We train neural network policies with DSS-GP-UCB and competing algorithms. We consider a sparse reward scenario where reward feedback is given every $S$ environment interactions for varying $S$. Table 1 shows that the performance of competing algorithms is severely degraded with sparse reward and DSS-GP-UCB outperforms competing approaches on most tasks with moderate or higher sparsity. Although intrinsic motivation (Singh et al., 2004; Zheng et al., 2018) has shown evidence in overcoming this limitation, we find that our approach outperforms competing approaches supported by intrinsic motivations at higher sparsity. This improvement

---

[9]We plot with respect to total environment interactions for l, and total policy evaluations for BO. See Appendix J, Appendix K, and Appendix L for alternate presentations of data more favorable to RL and MARL under which our conclusions still hold.

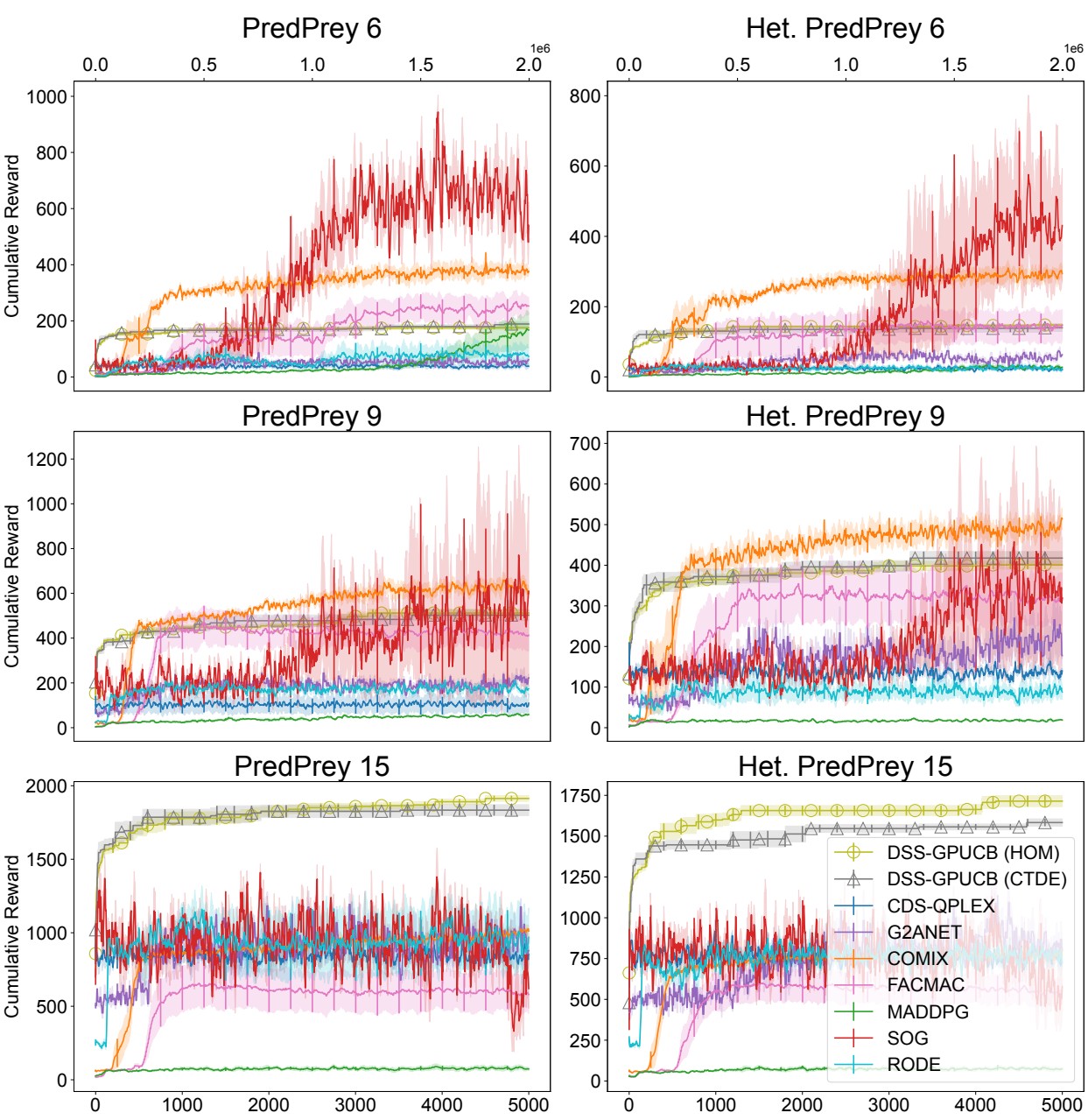

Figure 5: Scaling analysis. Training curves of DSS-GP-UCB and competitors with increasing number of agents. The left column shows PredPrey with 6, 9, and 15 agents. The right column shows Het, PredPrey with 6, 9, and 15 agents.

is important as sparse and malformed reward structure scenarios can occur in real-world tasks (Aubret et al., 2019). We repeat this validation in Appendix B with MARL algorithms in multi-agent settings and consider a delayed feedback setting with similar results.

## 5.4 Higher-order model Investigation

We examined policy for Multiagent Ant with 6 agents for the role based policy specialization. The policy modulation plots were generated by examining the PredPrey and Het. PredPrey environments respectively.

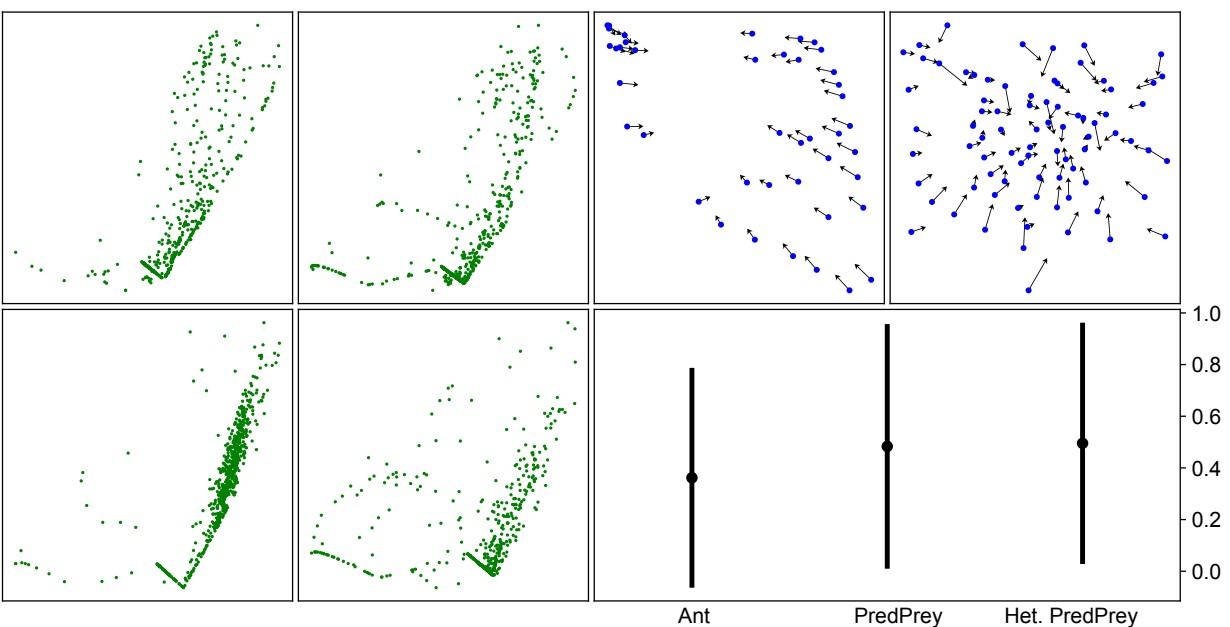

Figure 6: Left: Action distributions of different roles showing diversity in the Multiagent Ant environment with 6 agents. Right above: Policy modulation with role interaction in PredPrey and Het. PredPrey environment with 3 agents. Arrows represent change after message passing. These plots are visualizations of the two principal components after Principal Component Analysis. Right below: Mean connectivity ratio between agents and standard deviation in role interaction in Multiagent Ant with 6 agents, PredPrey with 3 agents, and Het. PredPrey with 3 agents.

In Fig. 6 we investigate the learned HOM policies. Our investigation shows that *role* is used to specialize agent policies while maintaining a common theme. *Role interaction* modulates the policy through graphical model inferences. Finally, role interactions are sparse, however noticeably higher for complex coordination tasks such as PredPrey.

### 5.5 Comparison with HDBO algorithms

We compare with several related work in HDBO. This is presented in Fig. 4, rightmost plots. We compare against these algorithms at optimizing our HOM policy. For more complex tasks that require role based interaction and coordination, our approach outperforms related work. TreeBO (Han et al., 2021) is also an additive decomposition approach to HDBO, but uses Gibbs sampling to learn the dependency structure. However, our approach of learning the structure through *Hessian-Awareness* outperforms this approach. Additional experimental results are deferred to Appendix B.

### 5.6 Drone delivery task

We design a drone delivery task that is well aligned with our motivation of considering policy search in *memory-constrained devices* on tasks with *unhelpful or noisy gradient information*. In this task, drones must maximize the throughput of deliveries while avoiding collisions and conserving fuel. This task is challenging as a positive reward through completing deliveries is rarely encountered (i.e., sparse rewards). However, agents often receive negative rewards due to collisions or running out of fuel. Thus, gradient-based approaches can easily fall into local minima and fail to find policies that complete deliveries.[10] We compare DSS-GP-UCB against competing approaches in Fig. 4, leftmost two plots. We observe that MARL based approaches fail to find a meaningfully rewarding policy in this setting, whereas our approach shows strong and compelling

---

[10]Further details on this task can be found in Appendix I.

performance. Furthermore, DSS-GP-UCB (MM) outperforms DSS-GP-UCB (CTDE) through leveraging roles and role interactions.

## 6 Conclusion

We have proposed a HOM policy along with an effective optimization algorithm, DSS-GP-UCB. Our HOM and DSS-GP-UCB are designed to offer strong performance in high coordination multi-agent tasks under sparse or malformed reward on memory-constrained devices. DSS-GP-UCB is a theoretically grounded approach to BO offering good regret bounds under reasonable assumptions. Our validation shows DSS-GP-UCB outperforms RL and MARL at optimizing neural network policies in malformed reward scenarios. Our HOM optimized with DSS-GP-UCB outperforms MARL approaches in high coordination multi-agent scenarios by leveraging the concepts of *role* and *role interaction*. Furthermore, we show through our drone delivery task, our approach outperforms MARL approaches in multi-agent coordination tasks with sparse reward. We make significant progress on high coordination multi-agent policy search by overcoming challenges posed by malformed reward and memory-constrained settings.

## Acknowledgements

We thank Jonathan Scarlett for pointing out a small mistake in our proof.

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

# A    Experimental Details

We used Trieste (Berkeley et al., 2022), Tensorflow (Abadi et al., 2015), and GPFLow (Matthews et al., 2017) to build our work and perform comparisons using MushroomRL (D'Eramo et al., 2021), MultiagentMuJoCo (de Witt et al., 2020), OpenAI Gym (Brockman et al., 2016), and Multi-agent Particle environment (Lowe et al., 2017). When comparing with related work, we used neural network policies of equivalent size. All of our tested policies are $< 500$ parameters, however the XL models are constructed using 3 layers of 400 neurons each.

To estimate the Hessian, we used the Hessian-Vector product approximation. We relaxed the discrete portions of our HOM policy into differentiable continuous approximation for this phase using the Sinkhorn-Knopp algorithm for the Role Assignment phase. For role interaction network connectivity, we used a sigmoid to create differentiable "soft" edges between each role. We pragmatically kept all detected edges in the Hessian while maintaining computational feasibility. We observed that our approach could support up to 1500 edges in the dependency graph prior to experiencing computational intractability. We used the Matern-$\frac{5}{2}$ as the base kernel in all our models.

## A.1    Ablation and Investigation

In the ablation, we perform experiments on MultiagentMuJoCo with environments Multiagent Ant with 6 segments, Multiagent Swimmer with 6 segments, Predator Prey with 3 predators, and Heterogeneous Predator Prey with 3 predators. In the Predator Prey environment, multiple predators must work together to capture faster and more agile prey. In Heterogeneous Predator Prey, each Predator has differing capabilities of speed and acceleration. This modification is challenging as a policy must not only coordinate between the Predators, but roles based specialization must be considered given the heterogeneous nature of each predator's capabilities.

To generate Fig. 6, we examined policy for Multiagent Ant with 6 agents for the role based policy specialization. The policy modulation plots were generated by examining the PredPrey and Het. PredPrey environments respectively.

## A.2    Comparison with MARL

For the MARL setting, we compare against MADDPG (Lowe et al., 2017), FACMAC (Peng et al., 2021), COMIX (Peng et al., 2021), RODE (Wang et al., 2021b) and CDS (Li et al., 2021) using QPLEX (Wang et al., 2021a) as a base algorithm. We also compare against Comm-MARL approaches SOG (Shao et al., 2022), and G2ANet (Liu et al., 2020). RODE and QPLEX are limited to discrete environments, thus we are unable to provide comparisons on continuous action space tasks such as Multiagent Ant or Multiagent Swimmer. All MARL environments were trained for $2,000,000$ timesteps. The neural network policies were 3-layers each with 15 neurons per layer, and were greater than or equal to the size of the compared HOM policy. For Actor-Critic approaches, we did not reduce the size or expressivity of the critic. All used hyperparameters and Algorithmic configurations were as advised by the authors of the work.

In the MARL setting we use Multiagent Ant, Multiagent Swimmer, Predator-Prey, Heterogeneous Predator-Prey. Multiagent Ant, and Multiagent Swimmer are MuJoCo locomotion tasks where each agent controls a segment of an Ant or Swimmer. Predator-Prey (PredPrey N) environment is a cooperative environment where N of agents work together to chase and capture prey agents. In Heterogeneous Predator Prey, each Predator has differing capabilities of speed and acceleration. This modification is challenging as a policy must not only coordinate between the Predators, but roles based specialization must be considered given the heterogeneous nature of each predator's capabilities. We also validated related work on the drone delivery task under which a drone swarm of N agents (Drone Delivery-N) must complete deliveries of varying distances while avoiding collisions and conserving fuel. The code of which is available in supplementary materials and will be open sourced.

We used batching (Picheny et al., 2022) in our comparisons with MARL to allow for a large number of iterations of BO. We used a batch size of 15 in our comparison experiments. In this setting, all MuJoCo environments

use the default epoch (total number of interactions with the environment for computing reward) length of 1000, for Predator-Prey environments, epoch length was 25, for Drone Delivery environment, epoch length was 150.

## A.3   RL and MARL under Malformed Reward

For single agent RL we compared against SAC (Haarnoja et al., 2018), PPO (Schulman et al., 2017), TD3 (Fujimoto et al., 2018), and DDPG (Lillicrap et al., 2015) as well as an algorithm using intrinsic motivation (Zheng et al., 2018). In single agent setting, we trained related work for $200,000$ timesteps. In the MARL setting, we trained for $2,000,000$ timesteps. In both single-agent setting and multi-agent setting all policy networks for both DSS-GP-UCB and related work was 3 layers of 10 neurons each. The tested environments were standard OpenAI Gym benchmarks of Ant, Hopper, Swimmer, and Walker2D.

In the MARL setting we compared against COVDN (Peng et al., 2021), COMIX, FACMAC, and MAD-DPG. Comparisons were not possible against other approaches as these do not support continuous action environments and are restricted to discrete action spaces.

For all environments and algorithms, we used the recommended hyperparameter settings as defined by the authors.

## A.4   Comparison with HDBO Algorithms

For this comparison, we compared with several related works in HDBO. We compared with TurBO (Eriksson et al., 2019b), Alebo (Letham et al., 2020), TreeBO (Han et al., 2021), LineBO (Kirschner et al., 2019), and a recent variant of BO for policy search, GIBO (Müller et al., 2021).

For computational efficiency, the epoch length for MuJoCo environments was reduced to 500.

## A.5   Drone Delivery Task

The experimental details follow that of comparisons with MARL.

## A.6   Compute

All experiments were performed on commodity CPU and GPUs. Each experimental setting took no more than 2 days to complete on a single GPU.

Table 2: Policy model sizes. Unfilled entries mean this environment was not considered during validation.

| | Ant-v3 | Hopper-v3 | Swimmer-v3 | Walker2d-v3 | Ant-v3 (MARL) | Hopper-v3 (MARL) | Swimmer-v3 (MARL) | Walker2d-v3 (MARL) |
|---|---|---|---|---|---|---|---|---|
| RL (Single Agent) | 478 | 263 | 222 | 356 | | | | |
| MARL (CTDE) | | | | | 310 | 267 | 267 | 353 |
| DSS-GP-UCB (Single Agent) | 478 | 263 | 222 | 356 | | | | |
| DSS-GP-UCB (CTDE) | | | | | 310 | 267 | 267 | 353 |

Table 3: Policy model sizes. Unfilled entries mean this environment was not considered during validation.

| | Multiagent-Swimmer 4 | Multiagent-Swimmer 8 | Multiagent-Swimmer 12 | Multiagent-Ant 8 | Multiagent-Ant 12 | Multiagent-Ant 16 | PredPrey 6 | PredPrey 9 | PredPrey 15 | Het. PredPrey 6 | Het. PredPrey 9 | Het. PredPrey 15 |
|---|---|---|---|---|---|---|---|---|---|---|---|---|
| MARL (CTDE) | 267 | 267 | 267 | 396 | 396 | 396 | 478 | 478 | 478 | 478 | 478 | 478 |
| DSS-GP-UCB (CTDE) | 267 | 267 | 267 | 396 | 396 | 396 | 478 | 478 | 478 | 478 | 478 | 478 |
| DSS-GP-UCB (MM) | 244 | 267 | 216 | 406 | 396 | 434 | 373 | 393 | 393 | 373 | 393 | 393 |

### A.7 Policy Sizes

We list the policy sizes of our models in Table 2 and 3.

Of note is in each environment, the compared against policy of RL or MARL is greater than or equal to in size vs. the policy optimized by DSS-GP-UCB.

### A.8 Hyperparameter for Higher-Order Model

For our HOM we utilized simple grid search in order to pick the hyperparameter settings. Overly large neural networks suffered from difficulty of optimization by BO, whereas, overly small neural networks suffered from performance difficulty on several environments. We found that neural networks of 3 layers, and 4 neurons each performed well across a wide number of tested environments.

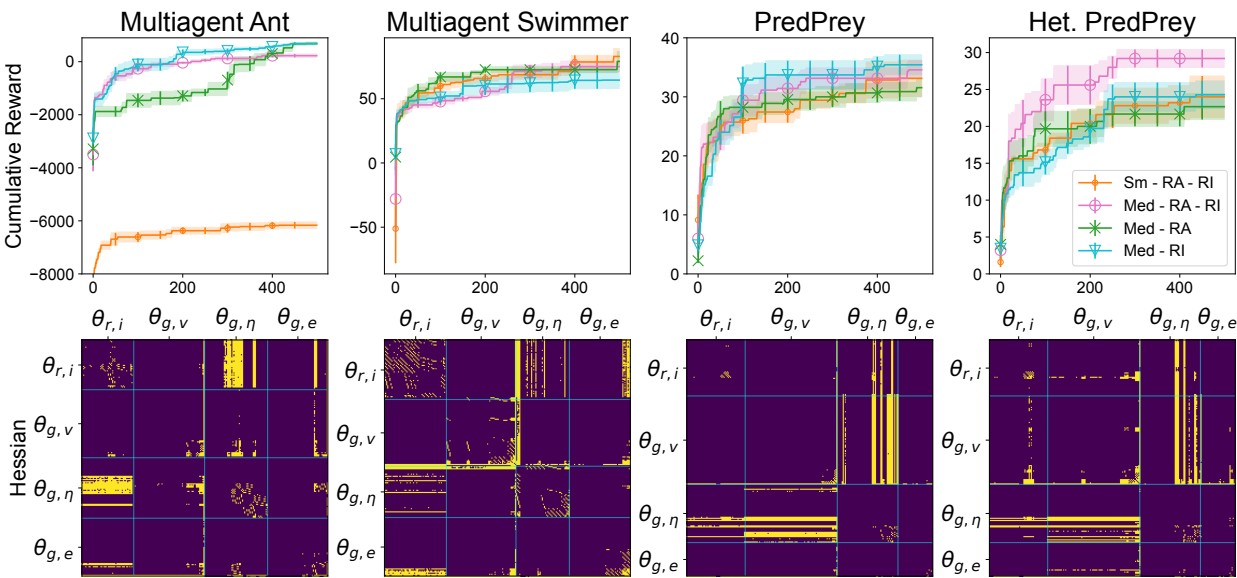

Figure 7: Ablation study. Training curves of DSS-GP-UCB and its ablated variants on different multi-agent environments.

## B  Additional Experiments

### B.1  Ablation

We present an expanded version of Fig. 3 in Fig. 7 including the ablation for Multiagent Swimmer. Multiagent Swimmer shows similar behavior as the simpler task Multiagent Ant, with stronger block-diagonal Hessian structure.

### B.2  Comparison with MARL

We present an expanded version of Fig. 5 in Fig. 8 including the results for Multiagent-Ant and Multiagent-Swimmer. We observe that in this relatively uncomplicated task not well-suited for our approach with dense reward, our HOM approach shows comparable performance to MARL approaches and far outperforms DSS-GP-UCB (CTDE). This shows the overall value of our HOM approach.

### B.3  RL and MARL under Malformed Reward

We present additional experiments under malformed reward for both RL and MARL. We formally define the Sparse reward scenario. Let $v(\theta) \triangleq \sum_{\Gamma=1}^{\hat{\Gamma}} r_\Gamma$ where the value of the policy is determined through $\hat{\Gamma}$ interactions with some unknown environment and each interaction is associated with the reward, $r_\Gamma$. Typically, RL algorithms observe the reward, $r_\Gamma$ after every interaction with the environment. We consider a sparse reward scenario where reward feedback is given every $S$ steps: $\tilde{r}_\Gamma^S \triangleq \sum_{\Gamma-S}^{\Gamma} r_\Gamma$ if $\Gamma \equiv 0 \mod S$ and 0 o.w. In addition to the sparse reward setting described earlier, we also consider the setting of delayed reward. The delayed reward scenario is defined: $\tilde{r}_\Gamma^D \triangleq r_{\Gamma-D}$ if $\Gamma > D$ and 0 o.w. Thus in the delayed reward scenario, feedback on an action taken is *delayed*. This scenario is important as it arises in long term planning tasks where the value of an action is not immediately clear, but rather is ascertained after significant delays. We present the complete table comparing related works in RL with DSS-GP-UCB in Table 4. As can be seen, similar to the Sparse reward scenarios, significant degradation can be observed across all tested RL algorithms with DSS-GP-UCB outperforming RL algorithms with moderate to severe amount of sparsity or delay. This degradation cannot be overcome by increasing the size of the policy, as we verify with the "XL" models which are orders of magnitude larger with 3 layers of 400 neurons.

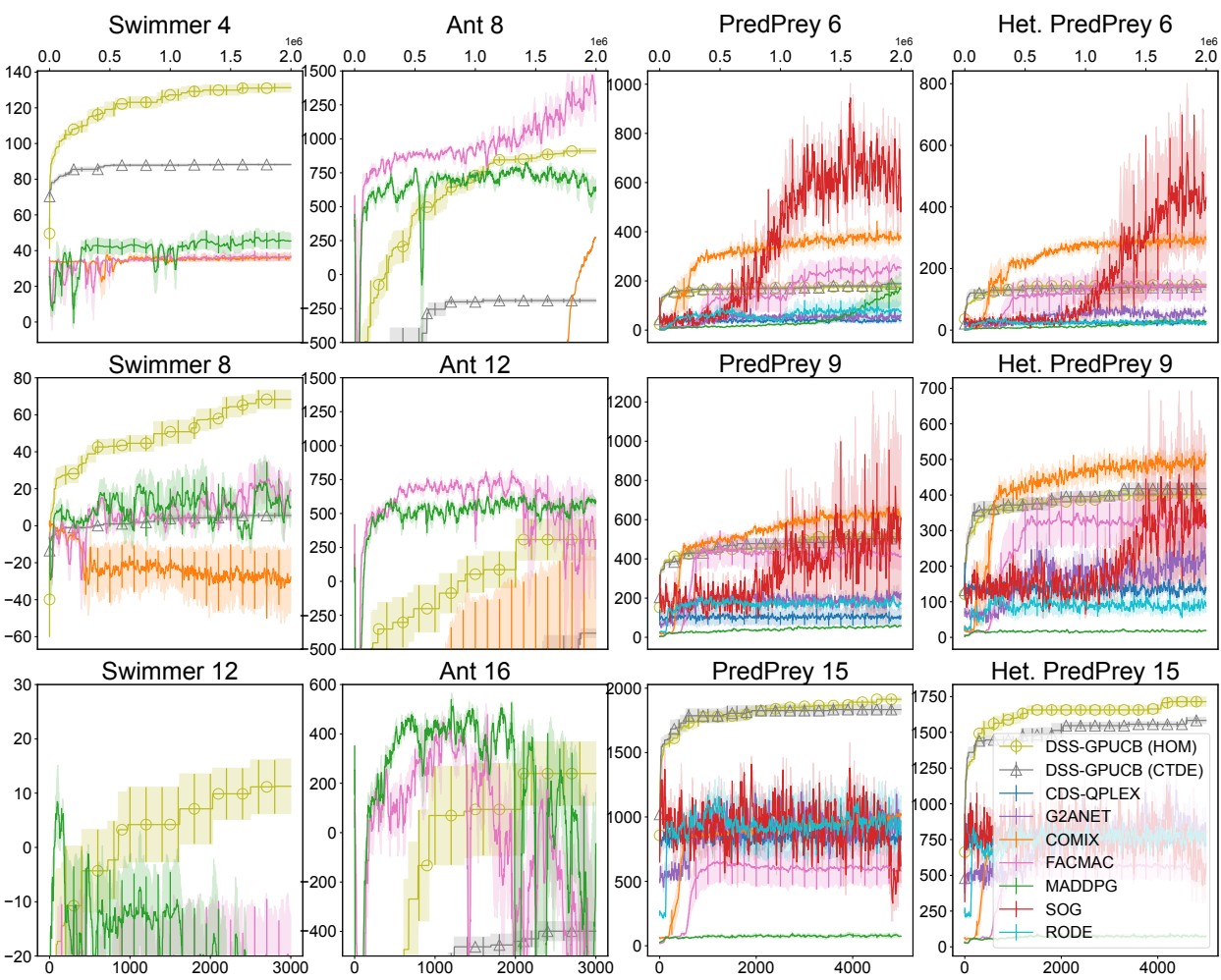

Figure 8: Comparison with MARL approaches with varying number of agents.

We repeat these experimental scenarios in the MARL setting with similar results in Table 5 where MARL approaches are compared against DSS-GP-UCB in the CTDE setting. Thus our validation shows that in both RL and MARL strong performance requires dense, informative feedback which may not be present outside of simulator settings. In these settings, our approach of optimizing small compact policies using DSS-GP-UCB outperforms related work in both RL and MARL.

Table 4: RL under sparse reward. Sparse $n$ refers to sparse reward. Delay $n$ refers to delayed reward. Averaged over 5 runs. Parenthesis indicate standard error.

| | Ant-v3 | | | | | Hopper-v3 | | | | | Swimmer-v3 | | | | | Walker2d-v3 | | | | |
|---|---|---|---|---|---|---|---|---|---|---|---|---|---|---|---|---|---|---|---|---|
| | DDPG | PPO | SAC | TD3 | Intrinsic | DDPG | PPO | SAC | TD3 | Intrinsic | DDPG | PPO | SAC | TD3 | Intrinsic | DDPG | PPO | SAC | TD3 | Intrinsic |
| Baseline | -90.77(318.36) | 1105.69(72.49) | 2045.24(477.29) | 2606.17(96.63) | 2144.00(56.19) | 604.20(153.01) | 1760.65(8.28) | 2775.66(396.61) | 1895.76(495.13) | 1734.00(152.46) | 44.45(7.03) | 121.38(5.25) | 56.73(1.86) | 48.78(0.48) | 1950.00(136.79) | 2203.80(177.40) | 892.81(208.68) | 4297.03(163.71) | 1064.46(190.97) | 210.00(223.94) |
| Sparse 2 | -3248.5(548.48) | 1007.80(93.20) | 2563.97(140.27) | 1407.46(238.70) | 1964.00(97.47) | 877.93(141.24) | 1567.14(113.01) | 3380.00(76.51) | 1579.84(627.28) | 2074.00(219.94) | 33.59(5.65) | 99.50(11.44) | 46.75(0.52) | 47.23(1.98) | 1758.80(254.49) | 1470.62(173.60) | 260.58(20.98) | 1673.46(704.37) | 1673.46(704.37) | 952.00(175.53) |
| Sparse 5 | -2867.97(362.46) | 961.31(103.83) | 711.56(87.40) | 762.64(38.90) | 1916.00(96.98) | 814.30(110.96) | 1616.78(330.77) | 3239.20(93.46) | 2290.67(560.51) | 1972.00(85.15) | 26.66(6.52) | 43.84(0.99) | 68.69(14.60) | 40.12(0.70) | 1856.00(237.63) | 961.30(231.98) | 227.49(344.78) | 697.93(678.48) | 2297.43(584.68) | 1924.00(213.72) |
| Sparse 20 | -2809.89(292.76) | 624.07(41.20) | 694.30(16.55) | 379.12(45.78) | 1838.00(389.72) | 783.95(175.83) | 1629.28(112.73) | 2535.17(368.16) | 1436.33(535.61) | 1537.20(336.19) | 19.12(12.07) | 54.63(9.12) | 37.78(2.06) | 37.03(2.63) | 2108.00(130.24) | 663.04(114.66) | 365.30(75.80) | 1010.63(490.13) | 276.56(69.79) | 1810.00(253.57) |
| Sparse 50 | -3067.37(116.57) | -67.43(33.33) | 663.28(21.26) | 253.66(149.78) | 1091.20(102.39) | 816.25(145.64) | 1010.73(165.81) | 1238.03(499.72) | 551.43(107.79) | 642.00(135.66) | 23.73(4.84) | 51.52(7.55) | 38.78(3.69) | 30.01(7.84) | 812.00(124.17) | 572.12(115.80) | 428.29(67.93) | 349.47(29.70) | 298.28(52.01) | 834.75(89.95) |
| Sparse 100 | -3323.43(43.74) | -4021.56(1893.85) | 679.30(23.68) | -115.43(287.63) | 450.40(51.04) | 988.36(15.38) | 324.51(53.61) | 260.52(31.38) | 342.48(36.19) | 406.80(51.77) | 9.64(9.63) | 21.09(4.97) | 27.98(6.35) | 30.10(1.95) | 376.60(80.28) | 523.89(114.07) | 205.93(18.11) | 200.16(26.90) | 147.22(24.45) | 480.60(75.14) |
| Sparse 200 | -3108.37(73.21) | -8167.98(2801.42) | -107.14(614.25) | -147.86(330.75) | 765.05(146.14) | 765.05(146.14) | 222.76(30.50) | 300.36(23.10) | 281.68(24.31) | 350.80(58.42) | -9.97(6.70) | 21.69(1.97) | 33.35(3.40) | 30.48(2.96) | 342.80(10.65) | 182.84(56.03) | 193.43(21.18) | 187.16(33.73) | 148.06(23.58) | 353.20(39.59) |
| Sparse XL 100 | -2887.23(182.60) | -5301.30(1979.27) | 448.67(118.92) | -730.72(730.12) | 429.28(141.31) | 608.79(116.39) | 481.72(68.89) | 328.95(10.91) | 188.00(67.16) | 570.00(59.04) | 9.56(8.54) | 31.47(1.56) | 36.70(2.86) | 50.99(1.73) | 473.60(67.95) | 867.98(65.29) | 222.13(23.32) | 250.27(12.96) | 95.01(43.31) | 421.80(31.56) |
| Sparse XL 200 | -2081.10(738.79) | -1108.40(2144.89) | 378.36(134.59) | -1931.39(563.91) | 373.40(28.56) | 373.40(28.56) | 361.34(60.88) | 331.69(17.64) | 217.74(0.89) | 331.00(27.99) | 9.87(3.57) | 55.52(3.31) | 36.61(12.84) | 35.21(1.31) | 414.40(20.72) | 843.38(70.49) | 206.66(31.57) | 228.33(32.14) | 270.81(40.87) | 339.20(39.73) |
| Lag 1 | -445.56(227.58) | 1028.21(103.66) | 866.70(5.83) | 860.53(14.72) | 1761.00(265.85) | 803.61(165.08) | 1812.64(219.72) | 3280.25(113.50) | 3377.79(48.14) | 1844.00(124.02) | 16.78(6.09) | 22.68(6.11) | 55.33(4.99) | 47.22(4.34) | 1980.00(106.70) | 2197.34(266.42) | 1180.54(362.37) | 3696.57(712.13) | 3846.52(125.83) | 1788.00(215.73) |
| Lag 5 | -2849.00(349.92) | 848.20(43.75) | 824.88(16.22) | 54.63(619.95) | 1718.00(331.74) | 692.46(145.45) | 1944.12(263.78) | 3385.05(99.86) | 632.85(47.74) | 1526.00(141.38) | 21.16(9.25) | 57.67(4.33) | 30.10(6.27) | 42.81(3.63) | 1910.00(165.43) | 1328.87(246.34) | 740.85(108.44) | 3786.54(1650.86) | 463.18(69.68) | 942.80(34.55) |
| Lag 20 | -2578.96(232.02) | 239.10(69.51) | 790.54(16.98) | 791.36(13.30) | 1155.60(240.31) | 907.45(108.70) | 815.54(169.38) | 596.78(111.18) | 261.85(28.64) | 1086.80(109.28) | 22.80(7.69) | 35.28(1.43) | 29.66(2.61) | 26.24(2.97) | 2018.00(245.36) | 388.86(89.26) | 209.38(33.69) | 2117.99(652.73) | 164.23(46.02) | 1726.00(66.79) |
| Lag 50 | -3093.43(184.86) | -176.74(92.67) | 691.43(22.64) | 640.74(150.22) | 257.60(39.28) | 1028.14(4.61) | 216.30(36.22) | 292.60(37.57) | 261.85(28.64) | 292.00(28.97) | 11.72(7.89) | 37.77(2.27) | 37.72(0.56) | 23.17(4.52) | 377.20(54.35) | 424.51(86.68) | 197.16(17.30) | 209.38(33.63) | 128.41(9.08) | 382.40(47.09) |
| Lag 100 | -2784.59(92.56) | -5569.67(2110.05) | 605.73(38.85) | 163.35(233.15) | 392.00(35.41) | 786.96(167.20) | 271.60(46.30) | 278.98(25.34) | 196.16(71.21) | 299.60(28.97) | 4.790(7.07) | 19.18(2.56) | 30.63(5.42) | 22.95(15.78) | 340.40(42.30) | 488.44(128.77) | 182.44(4.52) | 147.78(20.99) | 275.18(34.80) | 304.40(29.79) |
| Lag 200 | -2655.00(279.69) | -9419.68(2620.52) | 503.86(61.88) | -404.24(557.86) | 406.80(41.65) | 280.53(51.98) | 577.16(15.41) | 306.90(9.95) | 431.40(34.74) | 431.40(34.74) | 25.59(3.22) | 25.59(3.22) | 26.03(5.69) | 24.72(9.90) | 412.60(26.17) | 656.25(107.48) | 227.56(33.60) | 275.18(34.80) | 153.97(46.48) | 478.00(33.40) |
| Lag XL 100 | -2722.44(340.61) | -3140.81(1713.35) | 503.86(61.88) | -959.92(648.64) | 377.60(26.88) | 439.52(175.28) | 390.18(80.50) | 331.84(15.65) | 191.93(69.82) | 342.60(33.82) | 9.84(5.67) | 23.67(3.79) | 17.53(1.37) | 9.07(7.23) | 317.60(54.21) | 844.83(0.32) | 219.77(23.85) | 237.50(19.80) | 168.55(49.49) | 336.40(14.66) |
| DSS-GP-UCB | | 1147.32(36.88) | | | | | 1009.3(17.95) | | | | | 175.73(15.54) | | | | | 1008.90(11.95) | | | |

Table 5: MARL under sparse reward. Sparse $n$ refers to sparse reward. Delay $n$ refers to delayed reward. Averaged over 5 runs. Parenthesis indicate standard error.

| | Ant-v3 | | | | Hopper-v3 | | | | Swimmer-v3 | | | | Walker2d-v3 | | | |
|---|---|---|---|---|---|---|---|---|---|---|---|---|---|---|---|---|
| | COVDN | COMIX | MADDPG | FACMAC | COVDN | COMIX | MADDPG | FACMAC | COVDN | COMIX | MADDPG | FACMAC | COVDN | COMIX | MADDPG | FACMAC |
| Baseline | 970.50(5.96) | 959.20(2.83) | 982.58(54.19) | 909.37(40.08) | 38.92(0.06) | 38.88(0.06) | 305.58(107.48) | 638.80(245.03) | -0.05(7.75) | 13.43(0.56) | 11.21(0.52) | 14.06(0.11) | 249.21(9.65) | 275.85(10.04) | 280.30(22.16) | 543.71(180.78) |
| Sparse 5 | 877.09(20.95) | 906.38(7.48) | 912.61(12.48) | 882.94(46.35) | 138.07(72.33) | 39.84(0.21) | 320.74(123.44) | 38.76(0.13) | 10.13(2.16) | 7.10(4.77) | 14.19(0.81) | 13.05(0.36) | 341.89(51.92) | 260.58(29.98) | 236.37(28.88) | 267.70(118.68) |
| Sparse 50 | 242.69(252.04) | -212.74(75.21) | 772.29(41.13) | 799.71(11.76) | 909.88(71.73) | 38.88(0.16) | 362.66(264.39) | 172.37(109.04) | 8.49(2.52) | 10.50(2.19) | 12.06(0.34) | 12.10(1.01) | 190.92(8.87) | 121.51(67.31) | 227.60(31.76) | 287.36(67.07) |
| Sparse 100 | 403.99(208.18) | -756.61(201.61) | 490.19(62.22) | 564.73(75.80) | 443.95(170.88) | 27.74(9.52) | 47.47(7.08) | 38.78(0.06) | -3.44(3.28) | 1.12(3.02) | 10.11(3.29) | 14.14(1.19) | 106.24(66.51) | 194.91(17.35) | 159.24(14.53) | 444.41(215.77) |
| Sparse 200 | 65.04(89.82) | -197.26(64.03) | -632.05(967.04) | 584.23(28.34) | 687.82(249.11) | 38.78(0.02) | 50.74(9.86) | 37.10(1.36) | -0.15(3.39) | 14.43(0.73) | 14.43(0.73) | 13.84(0.13) | 165.50(67.12) | 277.89(4.89) | 123.91(10.61) | 229.11(129.53) |
| Sparse XL 100 | 766.66(208.18) | 209.58(201.61) | 632.15(62.22) | 552.28(75.80) | 813.86(170.88) | 236.74(9.52) | 50.62(7.08) | 72.11(0.06) | 9.52(3.28) | 14.45(3.02) | 13.69(3.29) | 14.16(1.19) | 135.95(66.51) | 28.22(17.35) | 127.66(14.53) | 184.52(215.77) |
| Sparse XL 200 | 442.92(89.82) | 322.54(64.03) | 479.03(967.04) | 533.20(28.34) | 996.85(249.11) | 39.47(0.02) | 26.41(9.86) | 71.31(1.36) | 3.58(3.39) | 4.67(1.20) | 13.28(0.73) | 14.34(0.11) | 167.62(67.12) | 207.81(4.89) | 157.37(10.61) | 88.47(129.53) |
| Lag 1 | 656.69(5.96) | 426.57(2.83) | 901.14(54.19) | 835.77(40.08) | 55.16(0.06) | 38.90(0.06) | 113.16(107.48) | 38.94(245.03) | 1.99(7.75) | 9.73(0.56) | 13.60(0.52) | 11.76(0.13) | 413.26(9.65) | 318.73(10.04) | 312.03(22.16) | 357.73(180.78) |
| Lag 5 | 255.56(20.95) | 302.46(7.48) | 873.81(12.48) | 910.90(46.35) | 629.94(72.33) | 38.85(0.21) | 376.38(123.44) | 612.84(264.39) | 0.79(2.52) | 6.26(2.19) | 14.55(0.34) | 11.16(1.01) | 170.91(51.92) | 289.44(29.98) | 302.41(28.88) | 640.21(118.68) |
| Lag 50 | 112.82(208.18) | -272.78(75.21) | 870.20(41.13) | 885.37(11.76) | 383.76(71.73) | 15.95(0.16) | 275.65(9.52) | 358.68(109.04) | 10.41(3.28) | 3.49(3.02) | 13.67(3.29) | 11.60(0.19) | 114.41(66.51) | 83.99(17.35) | 145.98(14.53) | 102.84(215.77) |
| Lag 100 | 366.62(89.82) | 10.21(201.61) | 844.80(62.22) | 843.90(75.80) | 434.69(249.11) | 38.95(0.02) | 26.44(9.86) | 38.74(1.36) | 5.27(3.39) | 7.93(1.20) | 13.28(0.73) | 13.86(0.13) | 51.59(67.12) | 147.23(4.89) | 147.23(14.53) | 277.79(129.53) |
| Lag 200 | 869.35(208.18) | 292.96(201.61) | 721.18(62.22) | 814.25(75.80) | 1000.04(170.88) | 38.85(9.52) | 532.12(7.08) | 26.30(0.06) | 9.23(3.28) | 0.48(3.02) | 10.12(3.29) | 10.19(1.19) | 105.65(66.51) | 119.80(17.35) | 191.74(14.53) | 280.79(215.77) |
| Lag XL 100 | 657.07(89.82) | 260.13(64.03) | 611.73(967.04) | 801.94(28.34) | 692.92(249.11) | 38.85(0.02) | 38.78(9.86) | 26.27(1.36) | 5.62(3.39) | 6.48(1.20) | 13.57(0.73) | 5.56(0.13) | 56.72(67.12) | 227.96(4.89) | 122.66(10.61) | 222.77(129.53) |
| DSS-GP-UCB (CTDE) | | 988.44(3.39) | | | | 213.50(22.26) | | | | 31.95(1.3) | | | | 397.68(14.42) | | |

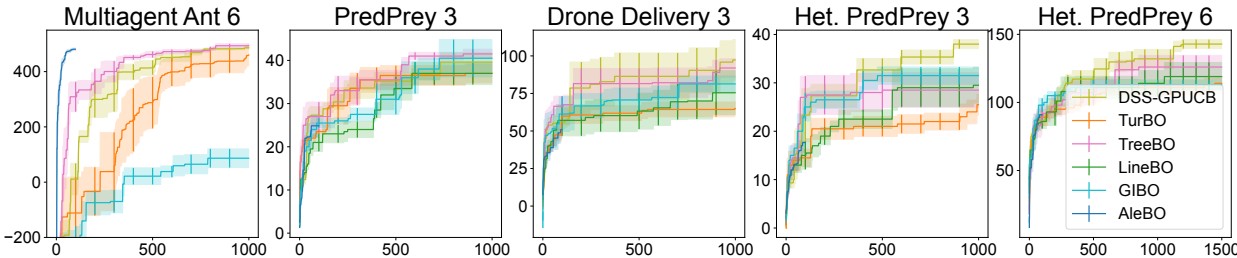

Figure 9: Comparison with BO algorithms. DSS-GP-UCB outperforms on complex multi-agent coordination tasks.

## B.4    Comparison with HDBO Algorithms

We compare with several related work in High-dimensional BO including TurBO (Eriksson et al., 2019b), AleBO (Letham et al., 2020), LineBO (Kirschner et al., 2019), TreeBO (Han et al., 2021), and GIBO (Müller et al., 2021). This is presented in Fig. 9. We experienced out-of-memory issues with AleBO after approximately 100 iterations, hence the AleBO plots are truncated. We compare against these algorithms at optimizing our HOM policy for solving various multi-agent policy search tasks. We validated on Multiagent Ant with 6 agents, PredPrey with 3 agents, Het. PredPrey with 3 agents, Drone Delivery with 3 agents, and also Het. PredPrey with 6 agents. We observe that these competing works offer competitive performance for simpler tasks such as Multiagent Ant and PredPrey with 3 agents. However for more complex tasks that require role based interaction and coordination, our approach outperforms related work. This is evidenced in Het. PredPrey 3, Het. PredPrey 6 as well as the Drone Delivery task with 3 agents.

Thus our validation shows that for simpler task, competing related works are able to optimize for simple policies of low underlying dimensionality. However, for more complex tasks which require sophisticated interaction using both Role and Role Interaction, related work is less capable of optimizing for strong policies *due to the complexity of the high-dimensional BO task*. In contrast, our work offers the capability of finding stronger policies for these complex tasks and scenarios.

Table 6: Summary of key notations.

| Notation | Description |
|---|---|
| $v$ | The objective function being optimized by Bayesian optimization |
| $\Theta$ | The domain for the objective function $v$ |
| $\theta_t$ | A point in the domain $\Theta$ that is picked at time $t$ |
| $\mu_T^k$ | The posterior mean (inferred after observations up to time $T-1$) at time $T$ using the kernel $k$ |
| $[\sigma_T^k]^2$ | The posterior variance at time $T$ using the kernel $k$ |
| $r(\theta_t)$ | The difference between the maxima of the function $v$ in domain $\Theta$, $v(\theta^*)$, and $v(\theta_t)$ |
| $R_T$ | The cumulative regret, $\sum_{t=1}^T r(\theta_t)$ |
| $\Theta^a$ | Dimension $a$ of the domain $D$ |
| $\mathcal{G}_d$ | A graph showing the dependencies between dimensions where edges exist between two dimensions if they are dependent |
| $V_d$ | In the graph indicated by $\mathcal{G}_d$ the set of dimensions corresponding to $\Theta$ |
| $E_d$ | In the graph indicated by $\mathcal{G}_d$ the set of edges corresponding to the dependencies between $\Theta$ |
| $\Theta^{(i)}$ | Collection of dimensions indicated by $(i)$ corresponding to a maximal clique in the graph $\mathcal{G}_d$ |
| $k^{\Theta^{(i)}}$ | A Gaussian process kernel correspond to the maximal clique $(i)$ |
| $k$ | The Gaussian process kernel for inference corresponding to the sum of $k^{\Theta^{(i)}}$ : $k \triangleq \sum_i k^{\Theta^{(i)}}$ |
| $v^{(i)}$ | Under the additive assumption, it is assumed that $v = \sum_i v^{(i)}$ where each $v^{(i)}$ is sampled from $k^{\Theta^{(i)}}$ |
| $\mathcal{U}(\Theta)$ | A uniform random distribution over the domain $\Theta$ |
| $H(\theta_{t,h})$ | A query to the Hessian at $\theta_{t,h}$ |
| $\widetilde{\mathcal{G}_d}$ | The graph corresponding to the detected dependency structure by querying the Hessian |
| Max-Cliques($\widetilde{\mathcal{G}_d}$) | A function computing the maximal cliques in the graph $\widetilde{\mathcal{G}_d}$ |
| $\mathbf{s}$ | The set of states of the cooperative multi-agent system where $\mathbf{s} \triangleq [\mathbf{s}^i]_{i=1,\ldots,n}$ and $i$ denotes the index of the agent |
| $\mathbf{a}$ | The set of actions taken by each agent where $\mathbf{a} \triangleq [\mathbf{a}^i]_{i=1,\ldots,n}$ and $i$ denotes the index of the agent |
| $\mathbf{s}^{\alpha(i)}$ | The state for agent $a$ taking on the role $\alpha(i)$ |
| $\mathbf{a}^{\alpha(i)}$ | The action taken by agent $a$ taking on the role $\alpha(i)$ |
| $\Lambda^{\theta_{r,i}}$ | An affinity function for taking on role $i$ where $r$ denotes it belonging to the part of the HOM for role assignment |
| $\Lambda^{\theta_{g,v}}$ | An affinity function determining whether an edge exists during the interaction of roles in the HOM policy |
| $M^{\theta_{g,\eta}}$ | The message passing function parameterized by $\theta_{g,\eta}$ for the role interaction message passing neural network |
| $U^{\theta_{g,e}}$ | The action update function parameterized by $\theta_{g,e}$ for the role interaction message passing neural network |

## C Table of Notations

Table 6 provides a summary of notations that are used frequently in paper.

# D    On the Applicability of Our Assumptions to RBF and Matern Kernel

We show that our assumption is satisfied by the RBF Kernel when $\Theta = [0,1]^D$, and is quasi-satisfied by the Matern$-\frac{5}{2}$ kernel. We also show that in the setting where $\Theta = [0,r]^D$ for some bounded $r$, our assumptions are quasi-satisfied as although these kernels may take on small negative values, these values decay exponentially with respect to the distance. These Lemmas show that our assumptions are reasonable.

**Lemma 1.** *Let $k\left(\theta, \theta'\right) \triangleq \exp(\frac{-d^2}{2})$ be the RBF kernel with $d \triangleq ||\theta - \theta'||$, then*

$$k^{\partial i \partial j}(\theta, \theta') = k\left(\theta, \theta'\right)\left(1 - (\theta^i - \theta'^i)^2\right)\left(1 - (\theta^j - \theta'^j)^2\right).$$

*Proof.* As shown in (Rasmussen & Williams, 2006) Section 9.4, the derivative of a Gaussian Process is also a Gaussian Process. Let $GP(0, k\left(\theta, \theta'\right))$ be the GP from which $f$ is sampled. This implies:

$$\frac{\partial f}{\partial \theta^a} \sim GP\left(0, \frac{\partial^2 k\left(\theta, \theta'\right)}{\partial \theta^a \partial \theta'^a}\right).$$

Applying this rule once more for the Hessian, we have:

$$\frac{\partial^2 f}{\partial \theta^b \theta^a} \sim GP\left(0, \frac{\partial^4 k\left(\theta, \theta'\right)}{\partial \theta^b \partial \theta'^b \partial \theta^a \partial \theta'^a}\right).$$

Given the above identities, we compute the partial derivatives for the RBF kernel:

$$\frac{\partial^2 k\left(\theta, \theta'\right)}{\partial \theta^a \partial \theta'^a} = \exp\left(-\frac{||\theta - \theta'||^2}{2}\right)\left(1 - (\theta^a - \theta'^a)^2\right).$$

Deriving once more we have:

$$\frac{\partial^4 k\left(\theta, \theta'\right)}{\partial \theta^b \partial \theta'^b \partial \theta^a \partial \theta'^a} = \exp\left(-\frac{||\theta - \theta'||^2}{2}\right)\left(1 - (\theta^a - \theta'^a)^2\right)\left(1 - (\theta^b - \theta'^b)^2\right).$$

This completes the proof noting that $k\left(\theta, \theta'\right) \triangleq \exp(\frac{-d^2}{2})$ with $d \triangleq ||\theta - \theta'||$. $\qquad\square$

**Corollary 1.** *Let $k\left(\theta, \theta'\right) \triangleq \exp(\frac{-d^2}{2})$, and $\theta, \theta' \in [0,1]^D$, then $k^{\partial i \partial j}(\theta, \theta') \geq 0$.*

*Proof.* The above is straightforward to see as $\exp\left(\cdot\right) \geq 0$ and with $\theta, \theta' \in [0,1]^D$ we have $\left(1 - (\theta^a - \theta'^a)^2\right) \geq 0$ $\left(1 - (\theta^b - \theta'^b)^2\right) \geq 0$. $\qquad\square$

**Corollary 2.** *Let $k\left(\theta, \theta'\right) \triangleq \exp(\frac{-d^2}{2})$, and $\theta, \theta' \in [0,r]^d$, then $k^{\partial i \partial j}(\theta, \theta') \geq c\exp(-d^2)$ for some constant $c$ dependent on $r$.*

*Proof.* The above is straightforward given the above Lemma. We note that although the RBF kernel may take on negative values in the domain $\Theta = [0,r]^d$, this values experience strong tail decay showing the quasi-satisfaction of our assumptions. $\qquad\square$

The above Lemma and Corollary shows that our assumptions are satisfied by the RBF Kernel when $\Theta = [0,1]^D$, and quasi satisfied when $\Theta = [0,r]^D$ after choosing a suitable $p_h$ and $\sigma_h^2$. We show how these assumptions are quasi-satisfied by the Matern-$\frac{5}{2}$ kernel.

**Lemma 2.** *Let $k\left(\theta,\theta'\right) \triangleq (1 + \sqrt{5}d + \frac{5}{3}d^2)\exp(-\sqrt{5}d)$ be the Matern-$\frac{5}{2}$ kernel with $d \triangleq ||\theta - \theta'||$, then with $d_i \triangleq \theta^i - {\theta'}^i$ we have*

$$k^{\partial i \partial j}(\theta,\theta') = \exp(-\sqrt{5}d)\left(\frac{5\sqrt{5}}{3} - \frac{25}{3d}d_i^2 - \frac{25}{3d}d_j^2 + \frac{25\sqrt{5}}{3d^2}d_i^2 d_j^2 + \frac{25}{3d^3}d_i^3 d_j^3\right).$$

*Proof.* Following the proof of Lemma 1, we state the partial derivatives of the Matern-$\frac{5}{2}$ kernel:

$$\frac{\partial^2 k\left(\theta,\theta'\right)}{\partial\theta^a \partial{\theta'}^a} = \exp\left(-\sqrt{5}||\theta - \theta'||\right)\left(\frac{5}{3} + \frac{5\sqrt{5}}{3}||\theta - \theta'|| - \frac{25}{3}(\theta^a - {\theta'}^a)^2\right).$$

Differentiating one more we have

$$\begin{aligned}
\frac{\partial^4 k\left(\theta,\theta'\right)}{\partial\theta^b \partial{\theta'}^b \partial\theta^a \partial{\theta'}^a} = &\exp\left(-\sqrt{5}||\theta - \theta'||\right) \\
&\left(\frac{5\sqrt{5}}{3} - \frac{25}{3d}(\theta^a - {\theta'}^a)^2 - \frac{25}{3d}(\theta^b - {\theta'}^b)^2 + \frac{25\sqrt{5}}{3d^2}(\theta^a - {\theta'}^a)^2(\theta^b - {\theta'}^b)^2 \right. \\
&\left. + \frac{25}{3d^3}(\theta^a - {\theta'}^a)^3(\theta^b - {\theta'}^b)^3\right).
\end{aligned}$$

This completes the proof noting that $d_i \triangleq \theta^i - {\theta'}^i$ and $d \triangleq ||\theta - \theta'||$. $\qquad\square$

**Corollary 3.** *Let $k\left(\theta,\theta'\right) \triangleq (1 + \sqrt{5}d + \frac{5}{3}d^2)\exp(-\sqrt{5}d)$ and $\theta,\theta' \in [0,1]^D$. Then $k^{\partial i \partial j}(\theta,\theta') \geq \exp(-\sqrt{5}d)\left(\frac{5\sqrt{5}}{3} - \frac{25}{3d} - \frac{25}{3d} - \frac{25}{3d^3}\right).$*

*Proof.* The above is an immediate consequence of Lemma 2 and noting that $||d_i|| \leq 1$. $\qquad\square$

**Corollary 4.** *Let $k\left(\theta,\theta'\right) \triangleq (1 + \sqrt{5}d + \frac{5}{3}d^2)\exp(-\sqrt{5}d)$ and $\theta,\theta' \in [0,r]^d$. Then $k^{\partial i \partial j}(\theta,\theta') \geq c\exp(-d)$ for some $c$ dependent on $r$.*

*Proof.* The above is an immediate consequence of Lemma 2 and noting that $||d_i|| \leq r$. $\qquad\square$

Although the above corollary shows that the Matern-$\frac{5}{2}$ kernel may take on negative values, we note that these values experience strong tail decay due to the presence of the $\exp\left(-\sqrt{5}d\right)$ term. Thus, the negative values are likely to be extremely small, thus quasi-satisfying our assumptions. In our experiments, we observed no shortcoming in using the Matern-$\frac{5}{2}$ kernel in DSS-GP-UCB.

# E   Proof of Proposition 1

We restate Proposition 1 for clarity.

**Proposition 1.** *Let $\mathcal{G}_d = (V_d, E_d)$ represent an additive dependency structure with respect to $v(\theta)$, then the following holds true: $\forall a, b \ \frac{\partial^2 v}{\partial \theta^a \partial \theta^b} \neq 0 \implies (\Theta^a, \Theta^b) \in E_d$ which is a consequence of $v$ formed through addition of independent sub-functions $v^{(i)}$, at least one of which must contain $\theta^a, \theta^b$ as parameters for $\frac{\partial^2 v}{\partial \theta^a \partial \theta^b} \neq 0$ which implies their connectivity within $E_d$.*

*Proof.* The above follows from the linearity of addition, which naturally implies a lack of curvature. In the multivariate case, this corresponds to zero or non-zero entries in the Hessian.

To be precise, we prove the contrapositive:

$$(\Theta^a, \Theta^b) \notin E_d \implies \frac{\partial^2 v}{\partial \theta^a \partial \theta^b} = 0.$$

Let $a, b$ be arbitrary dimensions with $(\Theta^a, \Theta^b) \notin E_d$. As a consequence of the definition of the dependency graph, $\nexists \Theta^{(i)}$ s.t. $\{\Theta^a, \Theta^b\} \subseteq \Theta^{(i)}$. That is, no subfunction $v^{(i)}$ takes both $\theta^a$ and $\theta^b$ as arguments.

By the linearity of the partial derivative, we see that:

$$\frac{\partial^2}{\partial \theta^a \partial \theta^b} v(\theta) = \frac{\partial^2}{\partial \theta^a \partial \theta^b} \sum_{i=1}^{M} v^{(i)}(\theta^{(i)}) = \sum_{i=1}^{M} \frac{\partial^2}{\partial \theta^a \partial \theta^b} v^{(i)}(\theta^{(i)}) = 0$$

where the last equality follows from no subfunction $v^{(i)}$ taking both $\theta^a$ and $\theta^b$ as arguments. $\qquad\square$

## F  Proof of Theorem 1

Our proof of Theorem 1 relies in being able to determine whether an edge does or does not exist in the dependency graph. To be able to do this, we examine the Hessian. As we have shown in Proposition 1, examining the Hessian answers this question. The challenge of Theorem 1 is detecting this dependency under noisy observations of the Hessian, as well as in domains where the variance of the second partial derivative is often zero, i.e., $k^{\partial i \partial j}(\theta, \theta') = 0$ with high probability. To overcome this challenge, we sample the Hessian multiple times to both find portions of the domain where $k^{\partial i \partial j}(\theta, \theta') \geq \sigma_h^2$, and also reduce the effect of the noise on learning the dependency structure. To proceed with the analysis, we first prove a helper lemma showing that if we can construct two Normal variables of sufficiently different variances, then it's possible to accurately determine which Normal variable has low, and high variance by taking a singular sample from each. This helper lemma will be used later to help determine edges in the dependency graph. As we shall soon show, If an edge exists, we are able to construct a Normal variable with high variance. Correspondingly, if an edge does not exist, we are able to construct a Normal variable with low variance.

**Lemma 3.** *Let $X_l \sim \mathcal{N}(0, \sigma_l^2)$ and $X_h \sim \mathcal{N}(0, \sigma_h^2)$ be two random univariate gaussian variables. For any $\delta \in (0,1)$, $\exists c_h$ s.t. $|X_l| \leq c_h \leq |X_h|$ with probability $1 - \delta$ when $\frac{\sigma_h^2}{\sigma_l^2} > \frac{8}{\delta^2} \log \frac{2}{\delta}$ and precisely when $\frac{\sigma_h \delta}{2} > c_h > \sigma_l \sqrt{2 \log \frac{2}{\delta}}$.*

*Proof.* First we note that $|X_l|$ and $|X_h|$ are Half-Normal random variables, with cumulative distribution function of $F_l(x) = \text{erf} \frac{x}{\sigma_l \sqrt{2}}$ and $F_h(x) = \text{erf} \frac{x}{\sigma_h \sqrt{2}}$ respectively. Thus to show that $|X_l| \leq \sigma_l \sqrt{2 \log \frac{2}{\delta}}$ and $|X_h| \geq \frac{\sigma_h \delta}{2}$ with high probability, we utilize well known bounds on the erf and erfc function. The proofs of the below can be found in several places, e.g., Chu (1955) and Ermolova & Häggman (2004) respectively.

$$\text{erf}\, x \leq \sqrt{1 - \exp -2x^2}\, ; \ \text{erfc}\, x \leq \exp -x^2.$$

Given the above, we show that $p(c_h \leq |X_l|) \leq \frac{\delta}{2}$ and $p(c_h \geq |X_h|) \leq \frac{\delta}{2}$ and utilizing the union bound completes the proof.

$$c_h > \sigma_l \sqrt{2 \log \frac{2}{\delta}} \implies c_h^2 > 2\sigma_l^2 \log \frac{2}{\delta} \implies \frac{c_h^2}{2\sigma_l^2} > -\log \frac{\delta}{2} \implies -\frac{c_h^2}{2\sigma_l^2} < \log \frac{\delta}{2}$$

$$\implies \exp -\frac{c_h^2}{2\sigma_l^2} \leq \frac{\delta}{2} \implies \text{erfc} \frac{c_h}{\sqrt{2}\sigma_l} < \frac{\delta}{2} \implies 1 - \text{erf} \frac{c_h}{\sqrt{2}\sigma_l} \geq 1 - \frac{\delta}{2} \implies F_l(c_h) \geq 1 - \frac{\delta}{2}$$

$$\implies p\left(c_h \leq |X_l|\right) < \frac{\delta}{2}.$$

Following a similar line of reasoning we have:

$$c_h < \frac{\sigma_h \delta}{2} \implies \frac{c_h^2}{\sigma_h^2} < \frac{\delta^2}{4} \implies \frac{-c_h^2}{\sigma_h^2} > -\frac{\delta^2}{4} \implies \frac{-c_h^2}{\sigma_h^2} > \log 1 - \frac{\epsilon^2}{4} \implies \exp -\frac{c_h^2}{\sigma_h^2} > 1 - \frac{\delta^2}{4}$$

$$\implies 1 - \exp -\frac{c_h^2}{\sigma_h^2} < \frac{\delta^2}{4} \implies \sqrt{1 - \exp -\frac{c_h^2}{\sigma_h^2}} < \frac{\delta}{2} \implies \text{erf} \frac{c_h}{\sigma_h \sqrt{2}} < \frac{\delta}{2} \implies F_h(c_h) < \frac{\delta}{2}$$

$$\implies p(c_h \geq |X_h|) < \frac{\delta}{2}.$$

Finally, to complete the proof, we show that the interval $(\sigma_l \sqrt{2 \log \frac{2}{\delta}}, \frac{\sigma_h \delta}{2})$ is not the empty set when $\frac{\sigma_h^2}{\sigma_l^2} > \frac{8}{\delta^2} \log \frac{2}{\delta}$.

$$\frac{\sigma_h^2}{\sigma_l^2} > \frac{8}{\delta^2} \log \frac{2}{\delta} \implies \frac{\sigma_h}{\sigma_l} > \frac{2\sqrt{2}}{\delta} \sqrt{\log \frac{2}{\delta}} \implies \frac{\sigma_h \delta}{2} > \sigma_l \sqrt{2 \log \frac{2}{\delta}}.$$

$\square$

We are now ready to prove Theorem 1.

**Theorem 1.** *Suppose[11] there exists $\sigma_h^2, p_h$ s.t. $\forall i, j$ $\mathbb{P}_{\theta \sim \mathcal{U}(\Theta)} \left[ k^{\partial i \partial j}(\theta, \theta) \geq \sigma_h^2 \right] \geq p_h$ and $\forall i, j, \theta, \theta'$ $k^{\partial i \partial j}(\theta, \theta') \geq 0$. Then for any $\delta_1, \delta_2 \in (0, 1)$ after $t \geq T_0$ steps of DSS-GP-UCB we have: $\bigcap_{i,j} P(\widetilde{E}_d^{i,j} = E_d^{i,j}) \geq 1 - \delta_1 - \delta_2$ when $T_0 = C_1 > \frac{16 D^2}{p_h \delta_1^2} \log \frac{2 D^2}{\delta_1} \frac{\sigma_n^2}{\sigma_h^2} + \frac{D^2}{2 \delta_2}$, $c_h \triangleq T_0 \sigma_n \sqrt{2 \log \frac{2 D^2}{\delta_1}}$.*

*Proof.* We prove the above for a single pair of variables, i.e., $k^{\partial i \partial j}$ and utilize the union bound to complete the proof. The first challenge to overcome is to sufficiently sample enough points in the domain such that we are able to find enough points $\theta \in \Theta$ where $k^{\partial i \partial j}(\theta, \theta) \geq \sigma_h^2$. To achieve this we sample $T_0$ different $\theta$ in the domain. After sampling $T_0$ points if there exists an edge between $\Theta^a$, and $\Theta^b$, then with probability $1 - \frac{\delta_2}{D^2}$ we have sampled $\frac{T_0 p_h}{2} - \frac{D^2}{2\delta_2}$ points where $k^{\partial i \partial j}(\theta, \theta) \geq \sigma_h^2$. To show the above we use bounds on the cumulative distribution of the Binomial distribution. A bound is given $T_0$ trials, with $p_h$ probability of success, the probability of having fewer than $s$ successes is upper bounded as follows:

$$\frac{1}{T_0 p_h - 2s}.$$

The above bound derives from the following well known tail bound Feller (1991):

$$P\left(S_n \leq s\right) \leq \frac{(T_0 - s)p_h}{(T_0 p_h - s)^2} \quad \text{if} \quad s \leq T_0 p_h$$

where $S_n$ denotes the number of successes. The above bound can be loosened by the following process:

$$\frac{(T_0 - s)p_h}{(T_0 p_h - s)^2} \leq \frac{T_0 p_h}{(T_0 p_h - s)^2} = \frac{T_0 p_h}{T_0^2 p_h^2 + s^2 - 2 T_0 p_h s} \leq \frac{T_0 p_h}{T_0^2 p_h^2 - 2 T_0 p_h s} =$$

$$\frac{T_0 p_h}{T_0 p_h (T_0 p_h - 2s)} = \frac{1}{T_0 p_h - 2s}$$

which yields the bound that we utilize. We note the above bound requires $s \leq \frac{T_0 p_h}{2} - \frac{1}{2}$, however if it is the case that $s \geq \frac{T_0 p_h}{2} - \frac{1}{2}$ then the worst case tail analysis is unnecessary since $\frac{T_0 p_h}{2} - \frac{1}{2} \geq \frac{T_0 p_h}{2} - \frac{D^2}{2\delta_2}$ and our results still hold.

Given the above, we use $\delta_2$ and derive:

$$\frac{1}{T_0 p_h - 2\left(\frac{T_0 p_h}{2} - \frac{D^2}{2\delta_2}\right)} \leq \frac{\delta_2}{D^2}.$$

Given the above, with at least $\frac{T_0 p_h}{2} - \frac{D^2}{2\delta_2}$ points where $k^{\partial i \partial j}(\cdot, \cdot) \geq \sigma_h^2$, as well as our assumption $k^{\partial i \partial j}(\theta, \theta) \geq 0$, we apply Bienaymé's identity which we restate for convenience:

$$\text{Var}\left[\sum_{\ell=1}^{C_1} h_{t,\ell}\right] = \sum_{\ell=1}^{C_1} \sum_{\ell'=1}^{C_1} \text{Cov}\left(h_{t,\ell}, h_{t,\ell'}\right).$$

---

[11]RBF kernel satisfies these assumptions when $\Theta = [0, 1]^D$.

Noting each of the $\frac{T_0 p_h}{2} - \frac{D^2}{2\delta_2}$ successes is sampled $C_1 = T_0$ times with $\text{Cov}\,(h_{t,\ell}, h_{t,\ell'}) \geq \sigma_h^2$ for each of the successes and $\text{Cov}\,(h_{t,\ell}, h_{t,\ell'}) \geq 0$ for all samples by our assumption. Applying Bienaymé's identity and the sum of (correlated) Normal variables is also a normal variable, we have $\text{Var}\left[\sum_{t=1}^{C_1} \sum_{\ell=1}^{C_1} h_{t,\ell}\right] \geq (\frac{T_0 p_h}{2} - \frac{D^2}{2\delta_2})T_0^2\sigma_h^2$. Compare this quantity with the variance if no edge exists between $\Theta^a$, and $\Theta^b$, where the variance results from i.i.d. noise: $\text{Var}\left[\sum_{t=1}^{T_0} \sum_{\ell=1}^{T_0} h_{t,\ell}\right] = T_0^2\sigma_n^2$. Comparing these two quantities, with an appropriately picked $c_h$ determines the edge between $\Theta^a$ and $\Theta^b$ using Lemma 3. By Lemma 3, letting $c_h \triangleq T_0 \sigma_n \sqrt{2\log\frac{2D^2}{\delta_1}}$ ensures that $p(h^{i,j} < c_h) < \frac{\delta_1}{D^2}$ if edge $E_d^{i,j}$ exists, and $p(h^{i,j} > c_h) < \frac{\delta_1}{D^2}$ if edge $E_d^{i,j}$ does not exist. Applying the union bound over $D^2$ pairs of variables completes the proof with $\bigcap_{i,j} P(\widetilde{E}_d^{i,j} = E_d^{i,j}) \geq 1 - \delta_1 - \delta_2$.

$\square$

# G   Proof of Theorem 2

Our proof of Theorem 2 is presented under the same setting and assumptions as the work of Srinivas et al. (2010).

To prove Theorem 2, we rely on several helper lemmas. The high-level sketch of the proof is to use the properties of Erdős-Rényi graph to bound both the *size of the maximal clique* as well as *the number of maximal cliques* with high probability. Once these two quantities are bounded, we are able to analyze the mutual information of the kernel constructed by *summing the kernels corresponding to the maximal cliques* of the sampled Erdős-Rényi graph as indicated in Assumption 1. Finally, once this mutual information is bounded, we use similar analysis as Srinivas et al. (2010) to complete the regret bound.

We begin by bounding the size of the maximal cliques.

**Lemma 4.** *Let $\mathcal{G}_d = (V_d, E_d)$ be sampled from a Erdős-Rényi model with probability $p_g$: $\mathcal{G}_d \sim G(D, p_g)$, then $\forall \delta \in (0,1)$ the largest clique of $\mathcal{G}_d$ is bounded above by*

$$|\text{Max-Clique}(\mathcal{G}_d)| \leq 2 \log_{\frac{1}{p_g}} |V_d| + 2 \sqrt{\log_{\frac{1}{p_g}} \frac{|V_d|}{\delta}} + 1$$

*with probability at least $1 - \delta$.*

*Proof.* The above relies on well known upper bounds on the maximal clique size on a graph sampled from an Erdős-Rényi model. As shown in (Bollobás & Erdös, 1976) and (Matula, 1976) the expected number of Cliques of size $k$, $\mathbb{E}[C_k]$ is given by:

$$\mathbb{E}[C_k] = \binom{|V_d|}{k} \frac{1}{p_g}^{-\binom{k}{2}} \leq |V_d|^k \frac{1}{p_g}^{-\frac{k(k-1)}{2}} = \frac{1}{p_g}^{\frac{k}{2}\left(2\log_{\frac{1}{p_g}} |V_d| - k + 1\right)}.$$

In the sequel, we omit the base of the log: $\frac{1}{p_g}$ for clarity. To bound the size of the maximal clique, we find a suitable $k$ such that $\mathbb{E}[C_k] \leq \frac{\delta}{n}$ and utilize the union bound over $[C_i]_{i=k,\ldots,n}$ where we have $|[C_i]_{i=k,\ldots,n}| \leq n$. Finally, we utilize Markov's inequality to complete the proof.

$$\text{Let } k = 2 \log |V_d| + 2 \sqrt{\log \frac{|V_d|}{\delta}} + 1.$$

We utilize the above bound on $\mathbb{E}[C_k]$.

$$\implies \frac{k}{2}\left(2\log_{\frac{1}{p_g}} |V_d| - k + 1\right) =$$

$$\left(\log|V_d| + \sqrt{\log \frac{n}{\delta}}\right)\left(2\log|V_d| - 2\log|V_d| - 2\sqrt{\log \frac{n}{\delta} + 1} + 1\right)$$

$$\leq -\log|V_d| - \log \frac{n}{\delta} + 1 \leq \log \frac{\delta}{n}$$

$$\implies \mathbb{E}[C_k] \leq \frac{1}{p_g}^{\log \frac{\delta}{n}} = \frac{\delta}{n}.$$

The proof is complete by noting that by Markov inequality, $p(C_k \geq 1) \leq \mathbb{E}[C_k]$ and taking the union bound over at most $n$ members of $[C_i]_{i=k,\ldots,n}$. $\qquad\square$

Next, we bound the total number of maximal cliques:

**Lemma 5.** *Let $\mathcal{G}_d = (V_d, E_d)$ be sampled from a Erdős-Rényi model with probability $p$: $\mathcal{G}_d \sim G(D, p_g)$, then $\forall \delta \in (0,1)$ the number of total maximal cliques in $\mathcal{G}_d$ is bounded above by*

$$\frac{1}{\delta} \sqrt{|V_d|^{\log_{\frac{1}{p_g}} |V_d| + 5}}$$

*with probability at least $1 - \delta$.*

*Proof.* We prove the above by bounding $\max_k C_k$ with high probability and noting that the number of maximal cliques is bounded by $\sum_k C_k \leq n \max_k C_k$ with high probability. To bound $\max C_k$, we first consider $\max_k \mathbb{E}[C_k]$.

$$\max_k \mathbb{E}[C_k] = \max_k \frac{1}{p_g}^{\frac{k}{2}\left(2\log_{\frac{1}{p_g}} |V_d| - k + 1\right)} = \frac{1}{p_g}^{\max_k \frac{k}{2}\left(2\log_{\frac{1}{p_g}} |V_d| - k + 1\right)}.$$

Taking the partial derivative of $\frac{k}{2}\left(2\log_{\frac{1}{p_g}} |V_d| - k + 1\right)$ with respect to $k$ we determine the maximum:

$$\arg\max_k \frac{k}{2}\left(2\log_{\frac{1}{p_g}} |V_d| - k + 1\right) = \log_{\frac{1}{p_g}} |V_d| + 1.$$

Thus we are able to bound:

$$\frac{\log_{\frac{1}{p_g}} |V_d| + 1}{2}\left(2\log_{\frac{1}{p_g}} |V_d| - \log_{\frac{1}{p_g}} |V_d| - 1 + 1\right) =$$

$$\frac{\log_{\frac{1}{p_g}} |V_d| + 1}{2}\left(\log_{\frac{1}{p_g}} |V_d|\right) = \frac{1}{2}\log^2_{\frac{1}{p_g}} |V_d| + \frac{1}{2}\log_{\frac{1}{p_g}} |V_d|$$

Which yields the bound:

$$\mathbb{E}[C_k] \leq \frac{1}{p_g}^{\frac{1}{2}\log^2_{\frac{1}{p_g}} |V_d| + \frac{1}{2}\log_{\frac{1}{p_g}} |V_d|} = \sqrt{|V_d|^{\log_{\frac{1}{p_g}} |V_d| + 1}}.$$

To complete the proof, we utilize Markov's inequality with $p\left(C_k \geq \frac{|V_d|}{\delta}\sqrt{|V_d|^{\log_{\frac{1}{p_g}} |V_d| + 1}}\right) \leq \frac{\delta}{|V_d|}$ and utilize the union bound over $n$ choices of $k$:

$$\sum_k C_k \leq \sum_k \frac{|V_d|}{\delta}\sqrt{|V_d|^{\log_{\frac{1}{p_g}} |V_d| + 1}} = \frac{1}{\delta}\sqrt{|V_d|^{\log_{\frac{1}{p_g}} |V_d| + 5}}$$

with probability $1 - \delta$. $\qquad\square$

Now that we have bounded both the number of cliques, as well as the sizes of the maximal cliques with high probability, we now consider the mutual information of the kernel constructed by summing the kernels corresponding to the maximal cliques of the dependency graph.

**Lemma 6.** *Define $I(\mathbf{y}_A; v) \triangleq H(\mathbf{y}_A) - H(\mathbf{y}_A \mid v)$ as the mutual information between $\mathbf{y}_A$ and $v$ with $H(\mathcal{N}(\mu, \Sigma)) \triangleq \frac{1}{2}\log|2\pi e \Sigma|$ as the entropy function. Define $\gamma_T^k \geq \max_{A \subset \Theta:|A|=T} I(\mathbf{y}_A; v)$ when $v \sim GP(0, k(\theta, \theta'))$. Let $[k_i]_{i=1,\dots,M}$ be arbitrary kernels defined on the domain $\Theta$ with upper bounds on mutual information $[\gamma_T^{k_i}]_{i=1,\dots,M}$, then the following holds true:*

$$\gamma_T^{\sum_i k_i} \leq M^2 \max [\gamma_T^{k_i}]_{i=1,\ldots,M}.$$

To prove the above, we first state Weyl's inequality for convenience:

**Lemma 7.** *Let $H, P \in \mathbb{R}^{n \times n}$ be two Hermitian matrices and consider the matrix $M = H + P$. Let $\mu_i, \nu_i, \rho_i, i = 1, \ldots, n$ be the eigenvalues of $M$, $H$, and $P$ respectively in decreasing order. Then, for all $i \geq r + s - 1$ we have*

$$\mu_i \leq \nu_r + \rho_s.$$

The above has an immediate Corollary as noted by Rolland et al. (2018):

**Corollary 5.** *Let $K_i \in \mathbb{R}^{n \times n}$ be Hermitian matrices for $i = 1, \ldots, M$ with $K \triangleq \sum_i^M K_i$. Let $[\lambda_\ell^{K_i}]_{\ell=1,\ldots,n}$ denote the eigenvalues of $K_i$ in decreasing order. Then for all $\ell \in \mathbb{N}_0$ such that $\ell M + 1 \leq n$ we have*

$$\lambda_{\ell M+1}^K \leq \sum_{i=1}^M \lambda_{\ell+1}^{K_i}.$$

We are now ready to prove Lemma 6 using Weyl's inequality and its corollary as a key tool.

*Proof.* Given the definition of $I(\mathbf{y}_A; v) \triangleq \frac{1}{2}\log|\mathbf{I} + \sigma^{-2}\mathbf{K}_A^k|$ (Srinivas et al., 2010) we bound the eigenvalues of $M\mathbf{I} + \sigma^{-2}\sum_i^M \mathbf{K}_A^{k_i}$ using the eigenvalues of $[I + \sigma^{-2}\mathbf{K}_A^{k_i}]_{i=1,\ldots,M}$ where $k \triangleq \sum_{i=1}^M k_i$. Using the above Corollary we see that:

$$\lambda_\ell^{M\mathbf{I}+\sigma^{-2}K} \leq \sum_{i=1}^M \lambda_{\lceil \frac{\ell}{M} \rceil}^{I+\sigma^{-2}K_i}.$$

Given the above, we see that $M^2 \max[\gamma_T^{k_i}]_{i=1,\ldots,M} \geq \frac{1}{2}\log|\mathbf{I} + \sigma^{-2}\mathbf{K}_A^k|$ as $\sum_i^M M\gamma_T^{k_i} \geq \frac{1}{2}\log|M\mathbf{I} + \sigma^{-2}\sum_i^M \mathbf{K}_A^{k_i}|$.

$\square$

Finally, we require an additional helper lemma to bound the supremum and infimum of a function sampled from a GP. This helper lemma helps bound the regret during the first phase of DSS-GP-UCB where we randomly sample the Hessian over the domain.

**Lemma 8.** *Let $k(\theta, \theta')$ be four times differentiable on the continuous domain $\Theta \triangleq [0, r]^D$ for some bounded $r$ (i.e., compact and convex) with $f \sim GP(0, k(\theta, \theta'))$ then for all $\delta \in (0, 1)$ the following holds true:*

$$\sup_{\theta \in [0,r]^D} f \leq c_b \sqrt{D \log \delta^{-1}} = \mathcal{O}\left(\sqrt{D \log \delta^{-1}}\right).$$

$$\inf_{\theta \in [0,r]^D} f \geq -c_b \sqrt{D \log \delta^{-1}} = \Omega\left(-\sqrt{D \log \delta^{-1}}\right).$$

*for some constant $c_b$ dependent on $\delta$ and $r$, with probability $1 - \delta$.*

*Proof.* The proof of the above is contained in Srinivas et al. (2010) Lemma 5.8. We restate the key parts of the proof herein. In the setting of Srinivas et al. (2010) there exist constants $a, b_i$, such that

$$P\left[\sup_{\theta \in \Theta} \left|\frac{\partial v}{\partial \theta_i}\right| > L\right] \leq ae^{-b_i L^2}.$$

Letting $L = [\log(Da2/\delta)/\min_i b_i]^{1/2}$, we have that $ae^{-b_i L^2} \leq \delta/(2D)$ for all $i = 1, \ldots, D$, so that for $K_1 = D^{1/2}L$ by the mean value theorem, we have

$$P\left[|v(\theta) - v(\theta')| \leq K_1||\theta - \theta'|| \ \ \forall \ \theta, \theta' \in \Theta\right] \geq 1 - \delta/2.$$

$\square$

The proof is complete by noting that $K_1 = \mathcal{O}((\log \delta^{-1})^{1/2})$ and picking an appropriate $c_b$ dependent on $\delta$ and $r$.

We are now ready to prove Theorem 2.

**Theorem 2.** *Let $k$ be the kernel as in Assumption 1, and Theorem 1. Let $\gamma_T^k(d) : \mathbb{N} \to \mathbb{R}$ be a monotonically increasing upper bound function on the* mutual information *of kernel $k$ taking $d$ arguments. The cumulative regret of* DSS-GP-UCB *is bounded with high probability as follows:*

$$R_T = \widetilde{\mathcal{O}}\left(\sqrt{T\beta_T D^{\log D+5}\gamma_T^k(4\log D + c_\gamma)}\right) \tag{4}$$

*where $c_\gamma$ is an appropriately picked constant and the base of the logarithm is $\frac{1}{p_g}$.*

We restate the above theorem with more precision:

**Theorem 2.** *Let $k$ be the kernel as in Assumption 1, and Theorem 1 and for some constants $a, b$,*

$$P\left[\sup_{\theta \in \Theta}\left|\frac{\partial v}{\partial \theta_i}\right| > L\right] \leq ae^{-(L/b)^2}, i = 1, \ldots, D.$$

*Let $\gamma_T^k(d) : \mathbb{N} \to \mathbb{R}$ be a monotonically increasing upper bound function on the mutual information of kernel $k$ taking $d$ arguments. Let $k(\theta, \theta')$ be four times differentiable on the continuous domain $\Theta \triangleq [0, r]^d$ for some bounded $r$ (i.e., compact and convex). For any $\delta_1, \delta_2, \delta_3, \delta_4, \delta_5, \delta_6 \in (0, 1)$. Let, $\tilde{t} \triangleq t - T_0 C_1$ and let*

$$\beta_t = 2\log(\tilde{t}^2 2\pi^2/3\delta_6^2) + 2D\log(\tilde{t}^2 Dbr\sqrt{\log(4Da/\delta_6)})$$

*The cumulative regret of* DSS-GP-UCB *is bounded:*

$$P\left[R_T \leq 2C_1^2 c_b\sqrt{D\log\delta_5^{-1}} + \sqrt{C_2 T \beta_T \gamma_T} + 2 \quad \forall T \geq 1\right] \geq 1 - \delta_1 - \delta_2 - \delta_3 - \delta_4 - \delta_5 - \delta_6$$

*when $C_1 = \frac{16D^2}{p_h\delta_1^2}\log\frac{2D^2}{\delta_1}\frac{\sigma_n^2}{\sigma_h^2} + \frac{D^2}{2\delta_2} + 1$, $C_2 = 8/\log(1 + \sigma^{-2})$, and $\gamma_T = \frac{1}{\delta_4^2}D^{\log_{1/p_g} D+5}\gamma_T^k\left(2\log_{1/p_g} D + 2\sqrt{\log_{1/p_g} D/\delta_3 + 1}\right)$ where $c_b$ is some constant dependent on $\delta_5$.*

*Proof.* The proof is a consequence of the helper lemmas and theorems we have proved. First we consider Phase 1 of DSS-GP-UCB where $t \leq T_0$. By Theorem 1, at most $T_0 C_1 = C_1^2$ queries will be made during Phase 1, and Lemma 8 indicates the maximum regret for any query. Consulting the respective Theorem and Lemma, we are able to bound the cumulative regret during Phase 1 by:

$$2C_1^2 c_b\sqrt{D\log\delta_5^{-1}} = \mathcal{O}(D^{4.5}\log^2 D).$$

Considering Phase 2, we utilize Lemma 4, Lemma 5, Lemma 6 to bound the mutual information of the sampled kernel with high probability. The number of cliques is given by:

$$\frac{1}{\delta_4}\sqrt{D^{\log_{1/p_g} D+5}}.$$

The size of the largest clique is given by:

$$2\log_{1/p_g} D + 2\sqrt{\log_{1/p_g} D/\delta_3 + 1} \leq 2(\log_{1/p_g} D + \log_{1/p_g} D/\delta_3 + 1) \leq 4\log_{1/p_g} D/\delta_3 + 2$$

Following Lemma 6, we see that

$$\gamma_T^{\sum_i k_i} \leq M^2 \max[\gamma_T^{k_i}]_{i=1,\ldots,M} \leq \frac{1}{\delta_4^2}D^{\log_{1/p_g} D+5}\gamma_T^k(4\log_{1/p_g} D/\delta_3 + 2).$$

Let $c_\gamma = 4\log_{\frac{1}{p_g}} 1/\delta_3 + 2$ which yields the following information on the mutual information:

$$\gamma_T^{\sum_i k_i} \leq D^{\log_{1/p_g} D+5}\gamma_T^k(4\log_{1/p_g} D + c_\gamma).$$

The proof is complete by leveraging the connection between mutual information and cumulative regret as shown by Srinivas et al. (2010) where $\widetilde{\mathcal{O}}$ is the same as $\mathcal{O}$ with the log factors suppressed. $\qquad\square$

# H   On the Surrogate Hessian, $\mathbf{H}_\pi$

In Section 4.5 we remarked that although we cannot observe $\mathbf{H}_v$, we can observe a surrogate Hessian, $\mathbf{H}_\pi$ which is related to $\mathbf{H}_v$ by the chain rule. We justify our choice here with showing how $\mathbf{H}_\pi$ is an important sub-component of $\mathbf{H}_v$ (Skorski, 2019). Although the reasoning we give is in one dimension, an analogous argument can be made in arbitrary dimensions using the chain rule for vector-valued functions yielding the Hessian tensor (Magalhães, 2020). We have $v : \Theta \to \mathbb{R}$ is a function of the policy $\pi$ and can be expressed as a composition of functions:

$$v : \Theta \to \mathbb{R} = \hat{v}\left(\pi\left(\theta\right)\right). \tag{5}$$

In the above we use $\pi\left(\theta\right)$ as shorthand for $\pi\left(\mathbf{s}^\alpha, \mathbf{a}^\alpha; \theta\right)$ with $\hat{v}$ representing some unknown function. Using the definition of the Hessian we have:

$$\mathbf{H}_v \triangleq \left[\frac{\partial^2 v}{\partial \theta^a \partial \theta^b}\right]_{a,b=1,\dots,D} = \left[\frac{\partial^2}{\partial \theta^a \partial \theta^b} \, \hat{v}\left(\pi\left(\theta\right)\right)\right]_{a,b=1,\dots,D}$$

Where the above identity follows from the definition of $v$ in Eq. 5. We can now apply chain rule to express:

$$\frac{\partial^2}{\partial \theta^a \partial \theta^b} \, \hat{v}\left(\pi\left(\theta\right)\right) = \underbrace{\left[\mathbf{H}_{\hat{v}}(\pi(\theta))\frac{\partial \pi}{\partial \theta^a}(\theta)\right] \cdot \frac{\partial \pi}{\partial \theta^b}(\theta)}_{r(\theta)} + \underbrace{\frac{\partial^2 \pi}{\partial \theta^a \partial \theta^b}(\theta)}_{\mathbf{H}_\pi(\theta)} \cdot \underbrace{\nabla \hat{v}(\pi(\theta))}_{g(\theta)} \tag{6}$$

As we see in the above as a consequence of the chain rule, $\frac{\partial^2 \pi}{\partial \theta^a \partial \theta^b}$ forms an important sub-component $\frac{\partial^2 v}{\partial \theta^a \partial \theta^b}$. Given the above, we can simplify the above in the following manner:

$$\mathbf{H}_v = r + \mathbf{H}_\pi \circ g$$

where $r, g$, and $\mathbf{H}_\pi$ arise from the corresponding highlighted terms in Eq. 6 with $r$ representing some unknown remainder term and $\circ$ representing the Hadamard product. Given the above, it is straightforward to see how $\mathbf{H}_\pi$ serves as a surrogate Hessian for $\mathbf{H}_v$. Indeed if $r \neq -\mathbf{H}_\pi \circ g$ and $g$ has no zero entries then $\mathbf{H}_\pi \neq 0 \implies \mathbf{H}_v \neq 0$. In our use case, we are most concerned with non-zero entries in the Hessian, $\mathbf{H}_v$, and the surrogate Hessian, $\mathbf{H}_\pi$ is well served for determining $\mathbf{H}_v \neq 0$ due to the above.

Since $\pi\left(\theta\right)$ is shorthand for $\pi\left(\mathbf{s}^\alpha, \mathbf{a}^\alpha; \theta\right)$, to approximate $\mathbf{H}_\pi$ we average $\mathbf{H}_{\pi\left(\mathbf{s}^\alpha, \mathbf{a}^\alpha; \theta\right)}$ over state action pairs, $\left(\mathbf{s}^\alpha, \mathbf{a}^\alpha\right)$ formed through interaction of the policy with the unknown task environment.

A possible avenue of overcoming this limitation is considering Hessian estimation through zero'th order queries. Several works along this direction have recently appeared using Finite Differences (Cheng et al., 2021), as well as Gaussian Processes (Müller et al., 2021). We consider removing this dependency on the surrogate Hessian for future work.

# I Drone Delivery Task

Our drone delivery task was inspired by recent research work in studying unique problems in drone delivery vehicle routing problems (Dorling et al., 2017).

Drones fly from delivery point to delivery point where completing a delivery gives a large amount of reward, but running out of fuel and collisions give a small amount of negative reward. After completing a delivery, the delivery point is randomly removed within the environment. A collision gives a small amount of negative reward and momentarily stops the drone. Completing a delivery refills the drone fuel and allows it to continue to make more deliveries. The amount of reward given increases quadratically with the distance of the delivery to highly reward long distance deliveries which require long term planning. To compound this requirement for long term planning, fuel consumption also dramatically increases at high velocities to encourage long-term fuel efficiency planning. In this complex scenario requiring long term planning, RL approaches can easily fall into local minima of completing short distance, low reward deliveries and fail to sufficiently explore (under sparse reward) policies which complete long distance deliveries with careful planning.

Implementation code of this task can be found in supplementary materials.

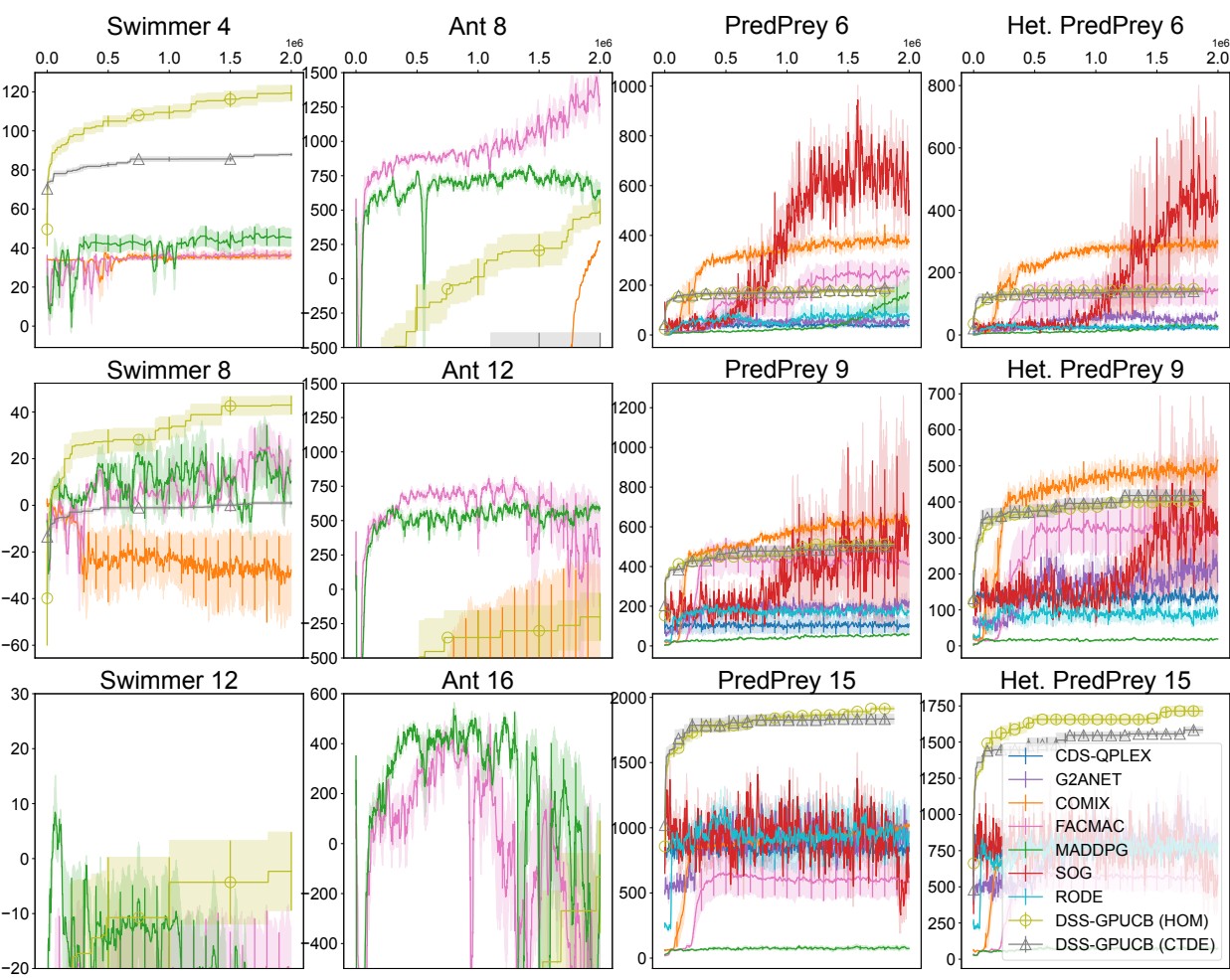

Figure 10: Comparison with MARL approaches with varying number of agents.

## J    Replot With Timesteps

We replot the relevant figures in Fig. 12 and Fig. 13 while maintaining total environment interactions as the singular independent variable. We note that there is no significant change to our conclusions as a consequence of this replotting. We also highlight that although total environment interactions is considered the important independent variable in RL and MARL, in BO typically the total evaluated policies is considered the more important independent variable as each evaluation is assumed to be costly.

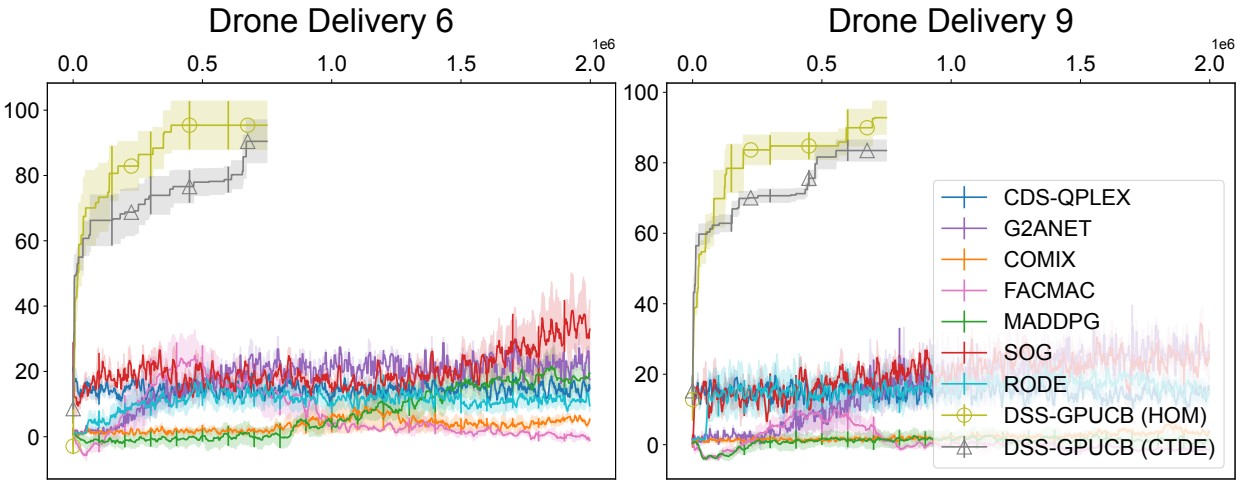

Figure 11: Comparison with MARL approaches on the drone delivery task.

## K    Replot With "best found policy so far" in RL

We replot the relevant figures in Fig. 12 and Fig. 13 where both BO and MARL approaches show the value of the "best found policy so far." We note that there is no significant change to our conclusions as a consequence of this replotting.

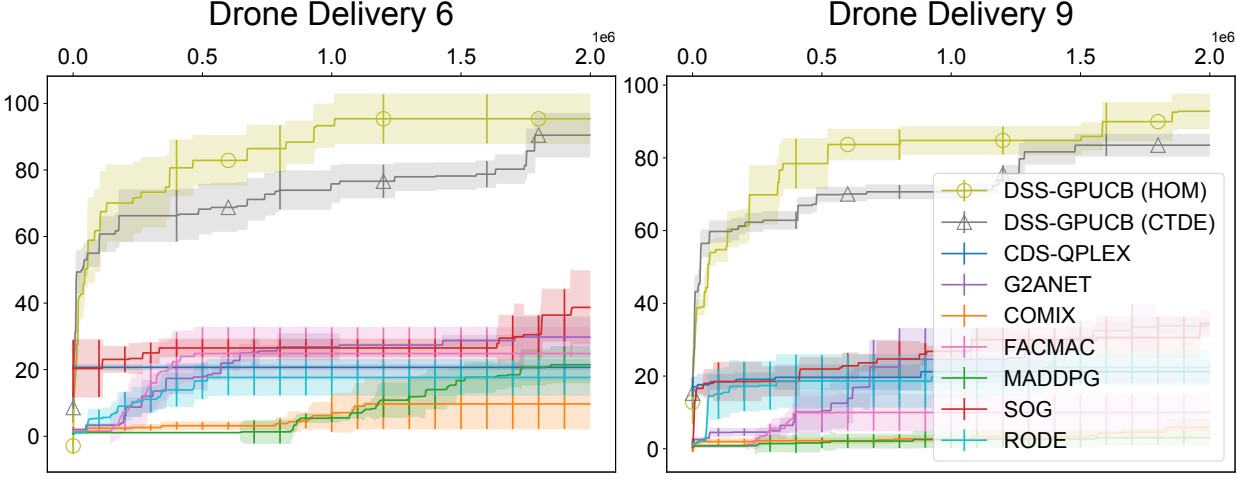

Figure 12: Comparison with MARL approaches with varying number of agents.

Figure 13: Comparison with MARL approaches on the drone delivery task.

## L    Tables With "best found policy so far" in RL

We generate new tables investigating RL and MARL under sparse or malformed reward. In Table 7 and 8 we show the value of the best found policy during the training process for RL and MARL. Our observations and conclusions remain the same where RL and MARL performance severely degrades under sparse and malformed reward and is often outperformed by our DSS-GP-UCB approach.

Table 7: RL under sparse reward. Sparse $n$ refers to sparse reward. Delay $n$ refers to delayed reward. Averaged over 5 runs. Parenthesis indicate standard error.

Table 8: MARL under sparse reward. Sparse $n$ refers to sparse reward. Delay $n$ refers to delayed reward. Averaged over 5 runs. Parenthesis indicate standard error.

