# OpenReview forum: "Dependency Structure Search Bayesian Optimization for Decision Making Models"
_TMLR — Accepted by TMLR_

### Review · Reviewer_LAWU · 2024-06-07

**Summary Of Contributions:**

This paper introduces Dependency Structure Search Bayesian Optimization (DSS-GP-UCB), a novel method that efficiently optimizes multi-agent decision-making models by leveraging role-based abstractions. This approach addresses the scalability issues in high-dimensional settings, making the optimization process more tractable and effective. Their method demonstrates strong empirical performance, particularly in environments with malformed or sparse rewards, and provides an improved regret bound under reasonable assumptions.

**Audience:**

Yes

**Claims And Evidence:**

Yes

**Requested Changes:**

* The formatting of Algorithms 1-4 needs to be revised, and additional descriptions should be provided for these algorithms.
* In the experimental results, the legend in Figure 4 should not obscure the Sparse reward curve.

**Strengths And Weaknesses:**

**Strengths:**

The content of this paper is substantial, and its structure is clear. The proposed DSS-GP-UCB method addresses the scalability issue in high-dimensional settings of complex decision-making models, which is a significant challenge for traditional Bayesian Optimization approaches. Moreover, the experiments presented in this paper are also comprehensive.

**Weaknesses:**

There are some confusing aspects in this paper, primarily in the algorithm description and theoretical guarantees. Regarding the algorithm description, Algorithm 4 is not described in sufficient detail, making it unfriendly to those unfamiliar with the GP-UCB algorithm. As for the theoretical guarantees, the use of O-notation twice in Theorem 2 is quite unusual.

---

> ### Author Response · Authors · 2024-07-19
> **Response to reviewer**
>
> There are some confusing aspects in this paper, primarily in the algorithm description and theoretical guarantees. Regarding the algorithm description, Algorithm 4 is not described in sufficient detail, making it unfriendly to those unfamiliar with the GP-UCB algorithm. As for the theoretical guarantees, the use of O-notation twice in Theorem 2 is quite unusual.
> - We have improved the text of the design section, and also described Algorithm 4 in more detail in both the text and in the algorithm text.
> - We have removed the O-notation within the O-notation both in the main text as well as in the proof in the Appendix. The resultant description is slightly less clean, as the previous description allowed us to suppress some additive constants.
>
>
> The formatting of Algorithms 1-4 needs to be revised, and additional descriptions should be provided for these algorithms.
> - We have improved the descriptions of Algorithm 1-4 in the text, as well as added more comments for Algorithm 4, which is the most complex described algorithm.
>
> In the experimental results, the legend in Figure 4 should not obscure the Sparse reward curve.
> - We have made the legend not obscure the curves. Although it looks slightly less refined as we compare our approach with numerous approaches in Figure 4.

---

### Review · Reviewer_kavf · 2024-06-07

**Summary Of Contributions:**

This paper presents DSS-GP-UCB, a role-based hierarchical Bayesian optimization (BO) method for cooperative multi-agent tasks. The authors introduce a role assignment and interaction model using role affinities and a graphical model, and a learning process for learning an additive decomposition of the observation dimensions. The authors state a regret bound, noting an improvement in the asymptotic complexity w.r.t. the number of dimensions. They also include experimental results on a few open multiagent environments and a synthetic drone delivery environment, comparing DSS-GP-UCB to a number of existing Bayesian Optimization and RL algorithms.

**Audience:**

Yes

**Claims And Evidence:**

No

**Requested Changes:**

Appendices don’t seem to be included.

Introduction: “This restriction requires the usage of large gradient-friendly policy representations”.  Do the authors have a specific sense of large they mean here, and an argument to support that? Otherwise, I’m not sure it requires large representations: I think it would be reasonable to say many RL algorithms -do- use larger NN to get empirically stronger performance, but not that they require large models. Smaller models – either small NN or linear weights on something like tile coding – can also have surprisingly competitive performance on some non-trivial problems.

“... and informative reward feedback from the environment.” This should be expanded to be more clear: without more context, observing the rewards for the trajectory an agent takes seems like a reasonably low requirement.

Section 4.3: “the number of interactions scales quadratically due to considering interactions between all pairs of agents.”  Isn’t pairwise interaction already a generalization, as interactions are possible between all agents? E.g., something doesn’t happen unless three (or more) agents coordinate on a decision.

“To generate graphical models of the above form, our HOM uses edge affinity functions… Edge affinity functions Lambda…”  Provide a (short description) of how: this is part of the contribution of this work, don’t just point at the algorithm.

Section 4.4: “Specifically Theta defeq Theta^1 x … x Theta^D for some dimensionality D, and Theta(i) … is of low dimensionality.” This definition seems to be general and allows for decomposition into multi-dimensional spaces. For the subsequent theory and bounds, are each of Theta^i assumed to be 1-dimensional spaces?

Section 4.5: “We utilize the surrogate Hessian in our work” Given the introductory motivation noting a limitation of MARL methods being they require gradient-friendly policy representations, it seems odd to see the proposed alternative computing any sort of Hessian.

Section 5: “interactions with the environment in l”  what is l?

Section 5.1: References for Multiagent Ant, PredPrey, and Mujoco

What exactly are the ablated agents? E.g., what does it mean to consider role interaction without role assignment?

Figure 4: label doesn’t seem to match. There are left, middle, and right, and all seem to be comparisons. Left plot is unlabeled. Middle plot is largely obscured by the labels.

What exactly is Sm compared to Med?

Section 5.2: What exactly is DSS-GP-UCB (CTDE)?

Section 5.3, Table 2: What is the setup? How long are agents trained? What does running DSS-GP-UCB mean if there are no rewards received on some steps? Given there is only a single line in Table 2 for DSS-GP-UCB, the authors are presumably saying there is absolutely no difference whatsoever between receiving rewards every step, or every 200 steps. How is this possible?

Figure 6: What are the axes?

**Strengths And Weaknesses:**

Strengths
The paper presents a BO setup and method with an improved regret bound. The experiments demonstrate that BO can be competitive with (or even better than) RL on at least some non-trivial environments.

Weaknesses
I understand there’s a lot of background information, and BO is not my general area of expertise, but I found the paper hard to follow. On the positive side, the authors do present examples and diagrams giving an overview in the spots where I would expect to see this, but the text / diagrams are following different conventions than I would expect or are assuming knowledge I don’t have.
For one concrete example, Figure 2 has state, Algorithm 2, and theta_r feeding into GEN to produce s^alpha(i). This, Alg 2, and theta_g are then feeding into GEN to produce pi. My immediate expectation from that diagram is that GEN is the same. At first impression, it’s also not clear what it means for Alg 1 to be an input of GEN. The text talks about a two stage process with an algorithm for the lower layer, so seeing a diagram with an algorithm as an input is not unexpected, even though it is in fact just the fixed step taken to generate the permutation. The diagram I would have expected to see is maybe something like

```
(s^1, …)      theta_r
    \            /
    Role Assignment
            |
       (s^alpha(1), …)
      /             \
Role Interaction    |
     |              |
  Graph N  theta_g  |
       \     |     /
           MPNN
             |
             pi
```

with the whole process being labeled as GEN.

At the point in 4.1 where Figure 2 is included, the choice to have “role” be entirely synonymous with permutation has not yet been introduced. Given an expectation of everyday use where we might expect to have a many to one mapping from index to some set of roles, having a permutation (s^alpha(1),...) is unexpected.

I found much of the explanation to have a similar problem: unexpected way of describing things, using things before describing them, or incomplete description – again noting that some of that incomplete description might just be assumed knowledge for someone more familiar with BO.

---

> ### Author Response · Authors · 2024-07-19
> **Response to reviewer**
>
> Introduction: “This restriction requires the usage of large gradient-friendly policy representations”. Do the authors have a specific sense of large they mean here, and an argument to support that? Otherwise, I’m not sure it requires large representations: I think it would be reasonable to say many RL algorithms -do- use larger NN to get empirically stronger performance, but not that they require large models. Smaller models – either small NN or linear weights on something like tile coding – can also have surprisingly competitive performance on some non-trivial problems.
> - The reviewer makes a good point. Although large policies are much more common in RL, it may not be something which is necessary and also is not the key point which distinguishes our approach from neural network approaches.
> - We have removed the wording “large” from this statement. The key distinguishing factor would be the presence of gradient friendly approaches which enable training through backpropagation.
> - We have instead emphasized IoT as the problem setting which enables BO to outperform RL approaches as small policies are needed. These small policies can be optimized using BO, whereas large policies cannot.
>
> “... and informative reward feedback from the environment.” This should be expanded to be more clear: without more context, observing the rewards for the trajectory an agent takes seems like a reasonably low requirement.
> - We have instead elected the more precise phrasing, “... dense reward, which allows reinforcement learning methods to infer a causal relationship between individual actions with their corresponding reward. This feedback may not be present in the scenario of sparse reward.”
>
> Section 4.3: “the number of interactions scales quadratically due to considering interactions between all pairs of agents.” Isn’t pairwise interaction already a generalization, as interactions are possible between all agents? E.g., something doesn’t happen unless three (or more) agents coordinate on a decision.
> - We have chosen the more accurate phrasing of, “can scale exponentially due to considering interactions between many agents”
> In addition, we have clarified around this topic as, the reviewer is correct that the pairwise interaction is a simplification offered by assumptions from graphical models.
>
> “To generate graphical models of the above form, our HOM uses edge affinity functions… Edge affinity functions Lambda…” Provide a (short description) of how: this is part of the contribution of this work, don’t just point at the algorithm.
> - We have improved the textual clarification in this portion, as well as for the other Algorithms described.
>
> Section 4.4: “Specifically Theta defeq Theta^1 x … x Theta^D for some dimensionality D, and Theta(i) … is of low dimensionality.” This definition seems to be general and allows for decomposition into multi-dimensional spaces. For the subsequent theory and bounds, are each of Theta^i assumed to be 1-dimensional spaces?
> - The definition is already general purpose, and each $\theta^{(i)}$ is allowed to be multi-dimensional. We have clarified the wording around this a bit better. Our assumption does require that each $\theta^{(i)}$ is at most on the order of O(log D) in size.
>
> Section 4.5: “We utilize the surrogate Hessian in our work” Given the introductory motivation noting a limitation of MARL methods being they require gradient-friendly policy representations, it seems odd to see the proposed alternative computing any sort of Hessian.
> - The reviewer is correct that this does seem a bit odd. We have clarified the wording around this question.
> - We note that the Hessian is only used to infer a simplification structure for policy search, and not used to train the model. Furthermore, our usage of the surrogate Hessian is the Hessian of the policy function, and not of the value function, which is certainly not considered.
>
> Section 5: “interactions with the environment in l” what is l?
> - Apologies, this was a small typo, it should read “interactions with the environment in RL”
>
> Section 5.1: References for Multiagent Ant, PredPrey, and Mujoco
> - We have added citations for these benchmarks.
>
> What exactly are the ablated agents? E.g., what does it mean to consider role interaction without role assignment?
> - We have improved the text around these experiments.
>
> Figure 4: label doesn’t seem to match. There are left, middle, and right, and all seem to be comparisons. Left plot is unlabeled. Middle plot is largely obscured by the labels.
> - We have added clarification here, the left 2 plots are grouped together, and the rightmost plot is separate.
> - We have also added cumulative reward on the y-axis for clarification.
>
> What exactly is Sm compared to Med?
> - We apologize for not clarifying. The small configuration is a very small neural network of 1 layer and 2 neurons per layer. We have clarified this in the main text.

---

> ### Comment · Reviewer_kavf · 2024-07-19
> **response to author response**
>
> (note that the responses to TYug and myself=kavf seem to be flipped)
>
> Thank you for the changes - they do notably improve the readability.
>
> I think the introductory text stating limitations of MARL methods has been improved. It does remain slightly odd to me that "gradient-friendly policy representation" is listed as a constraint, while using the Hessian of the policy is not. Similarly, the improvement in clarity going from "informative reward feedback ..." to "dense reward" is helpful, but the claim "requires ... dense reward" also still feels overly strong. "Usually performs better with"?\

---

> > ### Author Response · Authors · 2024-07-20
> > **Response**
> >
> > The reviewer makes a fair point. It is indeed true that this may be a difficult claim to defend, as we also rely on differentiability to enable some of our computations.
> >
> > Given that this is the case, we have softened the wording as requested, and double checked our assertions regarding gradient-based and differentiability within the paper to ensure that our key claim is in regard to the behavior of gradient based (or local) optimization methods compared to global optimization methods (i.e., BO).
> >
> > We have uploaded a new revision with these changes.

---

> ### Author Response · Authors · 2024-07-20
> **Response to reviewer (continued)**
>
> Section 5.2: What exactly is DSS-GP-UCB (CTDE)?
> - In the CTDE paradigm, both RI and RA are ablated reducing the policy model to a neural network which is identical across all agents.
> We have also clarified this in the main text.
>
> Section 5.3, Table 2: What is the setup? How long are agents trained? What does running DSS-GP-UCB mean if there are no rewards received on some steps? Given there is only a single line in Table 2 for DSS-GP-UCB, the authors are presumably saying there is absolutely no difference whatsoever between receiving rewards every step, or every 200 steps. How is this possible?
> - We do point out in the Appendix that all agents were trained with the given RL algorithms for 200,000 timesteps, which was more than sufficient to reach a stable policy for these simple environments.
> - We have also clarified the wording around how we intend to use DSS-GP-UCB to optimize a policy at the beginning of the design section, “In contrast to RL, which receives feedback on the reward of an action with every interaction, we treat $v(\theta)$ as an opaque function measuring the value of a policy. We utilize BO to optimize $\theta$ using solely the accumulated reward, $v(\theta)$, as feedback from the environment.”
> - The reviewer is correct that for DSS-GP-UCB there is no relationship with how often rewards are received, that is because from the perspective of BO, a $\theta$ is given to an opaque function $v$, and the reward for one epoch using the given $\theta$ is returned back to BO. Actually, the numbers presented represent BO receiving a reward every $1000$ steps, which is the epoch length for those experiments.
> - The reason why such an outcome is possible is BO attempts to take an optimization space (the policy function) and intelligently searches for the optima of the optimization space. The key assumption that BO makes is that the function to optimize is smoothly varying over the domain. We have found that this approach outperforms RL in the scenario of extremely sparse or adversarial reward.
>
> Figure 6: What are the axes?
> - For all the plots except the bottom right plot, there are no dimensions for the visualizations. We utilized PCA to reduce the action distribution to two dimensions, thus the x and y axis are dimensionless and can’t be labelled. For the bottom right plot, the axis is the ratio between the observed connectivity between roles divided by total number of possible pairwise connections. We have clarified this in the label.
> - For completeness, we have added a y-axis label for Figure 5.

---

### Review · Reviewer_TYug · 2024-07-07

**Summary Of Contributions:**

The authors utilize Bayesian optimization (BO) for high-dimensional multi-agent policy search (MAPS), as gradient-based methods can suffer from excessive computational requirements and sparse rewards, as well as the existence of local maxima. The authors mitigate the computational complexity of such an undertaking by relying on the abstractions of "role" and "role interaction"; they also assume an additive structure for the reward function, the exact structure of which they estimate using a surrogate for the parameters' Hessian.

**Audience:**

Yes

**Broader Impact Concerns:**

In my estimation the present work does not require a broader impact statement.

**Claims And Evidence:**

Yes

**Requested Changes:**

- Addressing the weaknesses mentioned above would strengthen the authors' arguments and exposition.
- The introduction is hard to follow and it takes a while to figure out the authors' contributions. E.g. the connection between the paragraph starting with "To optimize the HOM..." and the previous paragraph is not immediately clear. Figure 1 is not very helpful without the relevant context. I think a slightly longer introduction that includes a mathematical overview of the authors' method would be more helpful (Table 1 can be moved to Appendix), but I am happy with this structure as well if it does a better job of onboarding the reader.
- The concept of belief $\nu$ is not introduced.
- The letter $r$ is used to denote both the reward function and role.
- How are the cutoff constant $c_h$ determined?
- Please explicitly introduce that "Med" in "Med - RA - RI" refers to medium size.

**Strengths And Weaknesses:**

### Strengths
- The authors tackle an important problem that is likely to become more important as increased automation requires more lightweight and efficient solutions for MAPS.
- The authors present a promising solution that addresses the problems they associate with alternative methods. They provide empirical results as well as theoretical findings that support the usefulness of their method.

### Weaknesses
- The authors' method assumes the existence of a small number of roles that cover the roles of the agents. Further discussion into when this assumption is reasonable and when it is not is important to have in the main text. Their experiments indeed show that for some tasks it might be preferable to not have this component. The maximum number of agents they include in their experiments is 15, one would expect their method to be relevant in scenarios where num. roles << num. agents.
- A similar assumption is made for the value function. A better justification for this decision in realistic MAPS problems should be provided, and the resulting behavior when this assumption does not hold should be discussed.
- Hyperparameter selection for the two central assumptions described above is not described in detail. More guidance regarding the choice of number of roles and number of additive components should be provided.

---

> ### Author Response · Authors · 2024-07-19
> **Response to reviewer**
>
> The authors' method assumes the existence of a small number of roles that cover the roles of the agents. Further discussion into when this assumption is reasonable and when it is not is important to have in the main text. Their experiments indeed show that for some tasks it might be preferable to not have this component. The maximum number of agents they include in their experiments is 15, one would expect their method to be relevant in scenarios where num. roles << num. agents.
> - We address this concern in the common response.
>
> A similar assumption is made for the value function. A better justification for this decision in realistic MAPS problems should be provided, and the resulting behavior when this assumption does not hold should be discussed.
> - Could the reviewer clarify what assumption is meant? Is it the reliance of a surrogate Hessian?
>
> Hyperparameter selection for the two central assumptions described above is not described in detail. More guidance regarding the choice of number of roles and number of additive components should be provided.
> - We have added description regarding the selected hyperparameters, and how they were determined in Appendix A.8.
> - For our HOM we utilized simple grid search in order to pick the hyperparameter settings. Overly large neural networks suffered from difficulty of optimization by BO, whereas, overly small neural networks suffered from performance difficulty on several environments. We found that neural networks of 3 layers, and 4 neurons each performed well across a wide number of tested environments.
> - We also described how the hyperparameters were selected for DSS-GP-UCB in Appendix A.
>
> The introduction is hard to follow and it takes a while to figure out the authors' contributions. E.g. the connection between the paragraph starting with "To optimize the HOM..." and the previous paragraph is not immediately clear. Figure 1 is not very helpful without the relevant context. I think a slightly longer introduction that includes a mathematical overview of the authors' method would be more helpful (Table 1 can be moved to Appendix), but I am happy with this structure as well if it does a better job of onboarding the reader.
> - We address this concern in the common response.
>
> The concept of belief $v$ is not introduced.
> - We have removed the reference to belief, which is not necessary to understand the usage of GP. We have instead chosen the more neutral phrasing, "probability distribution."
>
> The letter $r$ is used to denote both the reward function and role.
> - We have used the letter $\rho$ for the reward function.
>
> How are the cutoff constant $c_h$ determined?
> - For empirical implementation and experiments, we kept all detected edges (i.e., $c_h = \epsilon$). Due to computational infeasibility, we were only able to experiment with at most $1500$ detected edges for our experiments prior to encountering out of memory issues on commodity GPUs. We detail this in Appendix A.
>
> Please explicitly introduce that "Med" in "Med - RA - RI" refers to medium size.
> - We have clarified this in the text.

---

> > ### Comment · Reviewer_TYug · 2024-08-12
> >
> > I thank the authors for their response, and believe that the modifications they made to the paper resulted in tangible improvements. My comment above was indeed regarding the additive decomposition of the value function and the corresponding estimation of the dependence structure using the surrogate Hessian, the authors' changes to Sections 4.4-4.5 helped address this concern.

---

> > > ### Author Response · Authors · 2024-08-14
> > > **Response to reviewer**
> > >
> > > We thank the reviewer for taking the time to read our responses, and our newer revision. We are pleased that our improvements to the writing and presentation of the work helped in understanding our contribution.
> > >
> > > We are also available to answer any further questions regarding our work, as well as clarifying any concerns.

---

### Author Response · Authors · 2024-07-19
**Official comment**

Dear reviewers,

We humbly thank all of you for your time and effort in reviewing the paper. In its current state, the paper demonstrates many strengths according to the reviewers which we highlight here:

- The authors appreciated the importance of our problem setting of resource constrained MAPS under sparse or malformed reward.
- The authors believed that our proposed solution is promising, which addresses the unique challenges in our problem setting. Moreover, the improved regret bound of DSS-GP-UCB as well as its superior performance over RL approaches in some environments contributes to the novelty of our work.
- The validation presented is comprehensive and detailed, which contributes to the strength of our work.

We have taken the time to prepare a new revision with the requested changes wherever possible, as well as help clarify some of the reviewer’s questions.

Some common weaknesses that reviewers remarked upon are as follows:

- Why permutation of agents into roles.
- - We made this design decision as it is a simple and straightforward solution to the role assignment problem, which is well studied. One advantage of this approach is it reduces the role assignment to the better studied permutation problem. In addition, the Hungarian algorithm for performing role assignment in this manner can be easily made differentiable by the Sinkhorn Algorithm which helps us extract the surrogate Hessian.
- - One possible avenue of further experiments is that of considering the scenario of fewer roles than agents. This is something we are open to looking into for further validation.
- Writing of the Introduction and Design section.
- - We have improved the writing of the introduction and design section, adding clarifying text and improving the logical structure of these sections.

We look forward to hearing back from the reviewers.

Sincerely,

The authors of Dependency Structure Search Bayesian Optimization for Decision Making Models

---

### Author Response · Authors · 2024-09-14
**Next steps.**

Dear reviewers,

We're hoping that the review process can move forward. Could you let us know if there are any steps we should take? I think hopefully the reviewers have had time to make their decisions as far as recommendation goes.

Sincerely,

The authors of Dependency Structure Search Bayesian Optimization for Decision Making Models.

---

### Decision · Action_Editor_nbix · 2024-09-23

**Recommendation:** Accept as is

**Comment:**

The reviewers all agreed that the paper tackles an important problem and makes a useful theoretical contribution.

There were some concerns on the clarity of the paper, which have been addressed by the authors. In the final discussion all reviewers agree on an accept decision.

**Audience:**

Yes

**Claims And Evidence:**

Yes